Corrected: Publisher correction

# Disease-relevant transcriptional signatures identified in individual smooth muscle cells from healthy mouse vessels

Lina Dobnikar[1,2], Annabel L. Taylor [2], Joel Chappell[2], Phoebe Oldach[1,6], Jennifer L. Harman[2], Erin Oerton [1,7], Elaine Dzierzak[3], Martin R. Bennett[2], Mikhail Spivakov [1,4,5] & Helle F. Jørgensen [2]

Vascular smooth muscle cells (VSMCs) show pronounced heterogeneity across and within vascular beds, with direct implications for their function in injury response and atherosclerosis. Here we combine single-cell transcriptomics with lineage tracing to examine VSMC heterogeneity in healthy mouse vessels. The transcriptional profiles of single VSMCs consistently reflect their region-specific developmental history and show heterogeneous expression of vascular disease-associated genes involved in inflammation, adhesion and migration. We detect a rare population of VSMC-lineage cells that express the multipotent progenitor marker Sca1, progressively downregulate contractile VSMC genes and upregulate genes associated with VSMC response to inflammation and growth factors. We find that Sca1 upregulation is a hallmark of VSMCs undergoing phenotypic switching in vitro and in vivo, and reveal an equivalent population of Sca1-positive VSMC-lineage cells in atherosclerotic plaques. Together, our analyses identify disease-relevant transcriptional signatures in VSMC-lineage cells in healthy blood vessels, with implications for disease susceptibility, diagnosis and prevention.

[1] Nuclear Dynamics Programme, Babraham Institute, Babraham Research Campus, Cambridge CB22 3AT, UK. [2] Division of Cardiovascular Medicine, University of Cambridge, Cambridge Biomedical Campus, Cambridge CB2 0QQ, UK. [3] MRC Centre for Inflammation Research, University of Edinburgh, Little France Crescent, Edinburgh EH16 4TJ, UK. [4] Functional Gene Control Group, Epigenetics Section, MRC London Institute of Medical Sciences, Du Cane Road, London W12 0NN, UK. [5] Institute of Clinical Sciences, Faculty of Medicine, Imperial College, Du Cane Road, London W12 0NN, UK. [6]Present address: Sir William Dunn School of Pathology, University of Oxford, South Parks Rd, Oxford OX1 3RE, UK. [7]Present address: Centre for Molecular Informatics, Department of Chemistry, University of Cambridge, Lensfield Road, Cambridge CB2 1EW, UK. These authors contributed equally: Lina Dobnikar, Annabel L. Taylor. These authors jointly supervised this work: Mikhail Spivakov, Helle F. Jørgensen. Correspondence and requests for materials should be addressed to M.S. (email: mikhail.spivakov@lms.mrc.ac.uk) or to H.F.J. (email: hfj22@cam.ac.uk)

Vascular smooth muscle cell (VSMC) accumulation is a hallmark of cardiovascular diseases such as atherosclerosis that causes heart attack and stroke[1]. VSMCs are found within the medial layer of large blood vessels, provide mechanical strength to the vessel and regulate vascular tone to control blood flow and blood pressure. VSMCs within healthy vessels are quiescent and characterised by the expression of contractile proteins such as aSMA (also known as ACTA2), Myocardin (MYOCD) and SM-MHC (also known as MYH11). However, VSMCs display remarkable phenotypic plasticity. When stimulated by injury or inflammation, VSMCs downregulate expression of the genes responsible for contractility and acquire a phenotype characterised by increased extracellular matrix production, migration and proliferation[2,3].

VSMC heterogeneity within and between different vascular regions with regard to morphology, growth characteristics and expression of specific candidate genes has been identified previously[4]. The observed cell-to-cell variation might result from different vascular structure and blood flow[5,6], as well as from the distinct developmental origin of VSMCs in different vascular beds[7]. It has therefore been hypothesised that VSMCs displaying different levels of plasticity co-exist within the healthy vessel wall[3] and might contribute to the non-random disease susceptibility of individual parts of the vasculature. We and others recently demonstrated that VSMC accumulation in atherosclerosis and after injury results from extensive clonal expansion of a small number of VSMCs[8–10]. This suggests that cells undergoing expansion were originally different from the general VSMC population in the healthy vessel wall, highlighting a possible functional significance of VSMC heterogeneity.

Single-cell RNA-sequencing (scRNA-seq) enables genome-wide profiling of individual cells[11,12] and is therefore an ideal methodology to detect cellular heterogeneity in an unbiased manner. Here we combine different scRNA-seq methodologies to delineate VSMC heterogeneity in healthy arteries and provide global insight into the nature of distinct cell subsets. We show that while the contractile VSMC signature is expressed relatively uniformly across most cells, there are pronounced differences in single-VSMC expression profiles between and within vascular beds for genes involved in cell adhesion, migration and inflammation. Combining scRNA-seq with VSMC lineage tracing, we reveal a rare subset of VSMC-lineage cells expressing Stem Cell Antigen 1 (Sca1, encoded by Ly6a, referred to below as Ly6a/Sca1). Sca1-positive cells show progressive downregulation of contractile VSMC genes and increased expression of genes associated with wound healing, migration and activation of growth factor signalling. We provide experimental evidence indicating that upregulation of Sca1 is a hallmark of VSMCs undergoing phenotypic switching in vivo and detect an equivalent Sca1-positive cell population within atherosclerotic plaques. These results suggest that Sca1 marks a plastic VSMC sub-population that gives rise to functionally distinct cells in disease.

## Results

### Single-cell transcriptomes of vascular smooth muscle cells.
To delineate VSMC heterogeneity in healthy tissue, we initially generated transcriptional profiles of individual cells using Fluidigm C1 technology. Single cells were isolated from the medial layer of mouse aortas by enzymatic digestion (Fig. 1a) and processed for sequencing. The scRNA-seq profiles of 143 cells from four experiments passed stringent quality control (83% of the analysed cells; Supplementary Fig. 1; see Methods for details). On average, 6704 genes were detected per cell.

All analysed cells showed consistently high expression of the contractile VSMC markers Myh11, Acta2 and Tagln. Lower-expressed marker genes Myocd, Smtn, Vcl and Cnn1 were detected in most cells, similar to what was observed for housekeeping genes (Fig. 1b). This indicates that all medial cells analysed express a contractile VSMC signature. Consistent with this conclusion, principal component analysis (PCA) demonstrated that all analysed single cells clustered tightly with VSMC control samples and away from adventitial control samples, both generated using the tube control protocol (Fig. 1c). Furthermore, the pooled single-cell VSMC expression profiles correlated with bulk RNA-seq data ($R^2 = 0.51$; Fig. 1d). As expected, the scRNA-seq profiles of ex vivo cells were clearly distinct from published profiles of cultured VSMCs[13] (Fig. 1e), which are known to recapitulate some aspects of VSMC phenotypic switching[14]. In particular, ex vivo cells expressed higher levels of contractile genes compared with the cultured cells (Supplementary Fig. 2). Taken together, these results confirm that the single-cell datasets we have generated are of sufficient coverage, specificity and overall quality to unambiguously identify the profiled cells as contractile VSMCs.

### Regional differences in VSMC gene expression.
We applied single-cell transcriptomics to gain insight into the previously observed VSMC heterogeneity between the athero-prone aortic arch (AA) and the more disease-resistant descending thoracic aorta (DT)[15,16]. AA and DT VSMCs differ in expression of a number of genes, including posterior Hox genes and other transcription factors[15]. These differences may arise from the distinct embryonic origins of VSMCs in the AA (neural crest) and DT (mesoderm) regions[7]. Alternatively, VSMCs in both regions could be heterogeneous with respect to these genes, with specific cell subsets represented in different proportions in the AA compared with the DT. These scenarios can be addressed directly with scRNA-seq. Prior to analysing regional expression differences at the single-cell level, we generated bulk RNA-seq profiles of VSMCs isolated from the medial layer of AA and DT to define robust gene expression signatures associated with VSMCs from these regions at the population level, which informed single-cell analysis (Fig. 2a). In total, 442 genes showed significant differential expression ($p < 0.01$, $\log_2$ fold change > 1), of which 386 genes were upregulated in the AA and 56 genes in the DT (Fig. 2b and Supplementary Data 1). RT-qPCR in independent, paired AA and DT samples verified differential expression of 6 out of 6 genes tested (Dcn, Lum, Pde1c, Gpc3, 3632451O06Rik and Hoxa7; Fig. 2c). Overall, 88/442 genes identified as region-specific by RNA-seq were previously found to be differentially expressed in AA and DT VSMCs by microarray analysis[15]. Notably, 44% of the microarray-identified genes were lowly expressed (normalised $\log_2$-transformed read count < 4) and their regional specificity was less reproducible by RNA-seq compared with genes expressed at higher levels (Supplementary Fig. 3). Similar to previous reports[15,16], the genes upregulated in the DT population compared with the AA included many developmental regulators, including 15 Hox genes corresponding to a posterior identity, and were significantly enriched for the gene ontology (GO) terms "anterior/posterior pattern specification" and "regionalisation and developmental process" (Supplementary Data 2a). In contrast, genes upregulated in AA VSMCs were enriched for terms associated with immune response, cell proliferation and migration (for example Lbp, Tgfbi and Mmp12; Fig. 2b and Supplementary Data 2b), consistent with the increased response of AA cells to mitogens and inflammatory cytokines[7,17]. Additionally, genes typically associated with the synthetic VSMC phenotype (Spp1, Pde1c)[3,18] were expressed at higher levels in the AA compared with the DT VSMCs (Fig. 2b). In summary, population-level analysis of VSMC region-specific transcription identified

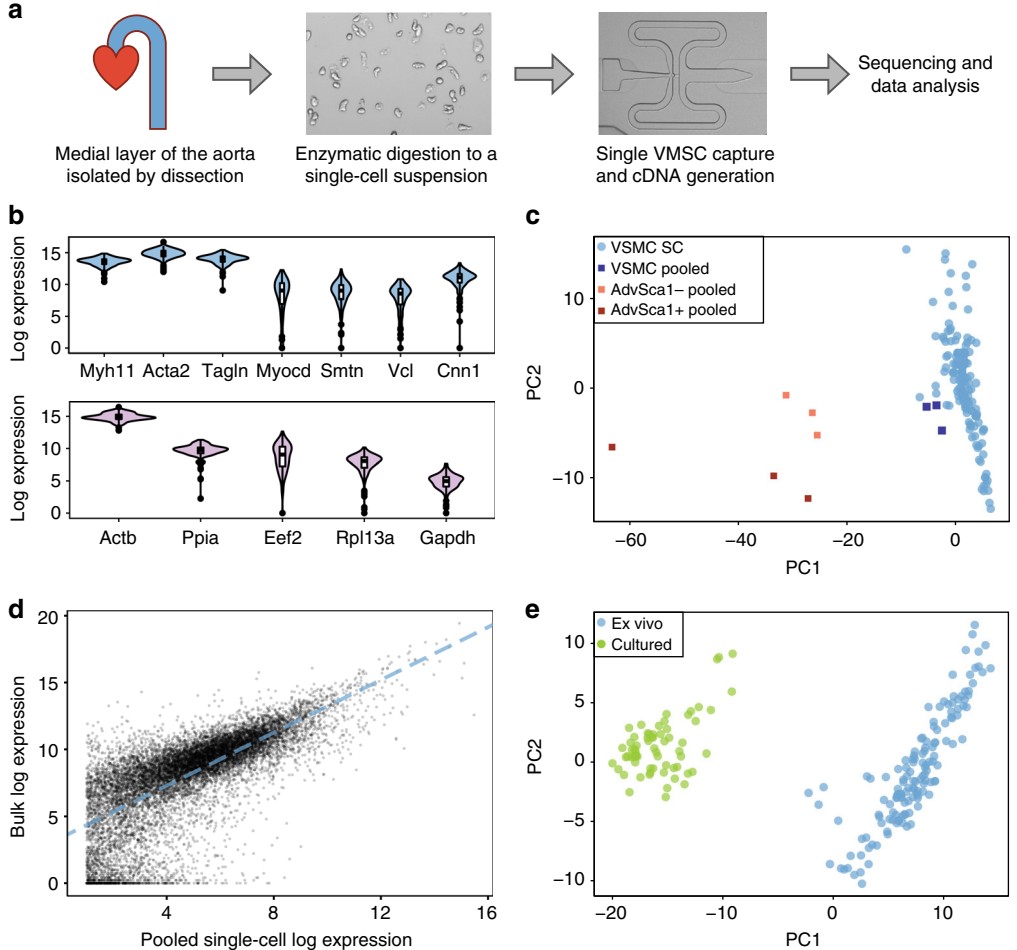

**Fig. 1** Single-cell RNA-seq analysis of vascular smooth muscle cells. **a** Schematic of the approach. Cells from the medial layer are enzymatically digested to obtain a single-cell suspension. Single-cell cDNA libraries are then generated, followed by sequencing and data analysis. **b** Violin plots showing the log$_2$-transformed normalised expression of VSMC marker genes across the profiled 143 cells (top), as well as of housekeeping genes with similar mean expression levels (lower panel). **c** Mapping of single-cell VSMC transcriptomes (light blue), as well as transcriptomes from control VSMC (Sca1–, dark blue) and adventitial (Adv) cell (Sca1–, orange; Sca1+, red) samples (tube controls) on a two-dimensional PCA space. **d** Dot plot showing the log$_2$-transformed read counts detected for each gene (black dots) when pooling across all single-cell samples versus the read counts detected with bulk RNA-seq. The dashed line shows a linear regression fit. **e** PCA plot summarising the single-cell expression profiles for ex vivo VSMCs (blue) and in vitro cultured VSMCs (green, data from Gene Expression Omnibus accession GSE79436, Adhikari et al.[13])

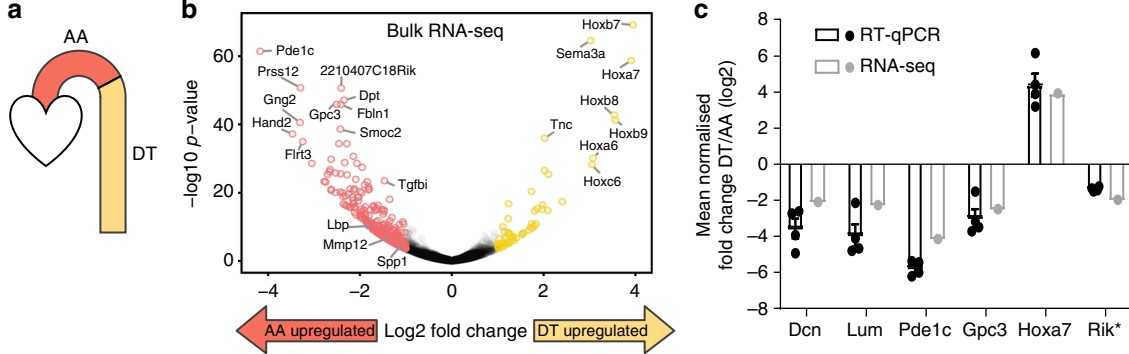

**Fig. 2** Bulk RNA-seq of VSMCs from the aortic arch and descending thoracic aorta. **a** Schematic representation of the aorta, indicating the aortic arch (AA) and descending thoracic aorta (DT). **b** Volcano plot showing significance (-log$_{10}$ p-value) versus relative gene expression in VSMCs from the DT versus the AA. Genes showing significant differences in expression (adjusted p-value < 0.01, log$_2$ fold change >1) are labelled in yellow (upregulated in DT) and red (upregulated in AA). Gene names for selected genes are indicated. Three independent samples were analysed for each region (AA and DT samples paired). **c** Relative expression (log$_2$(DT/AA)) of selected genes determined by RT-qPCR (black dots/bars, n = 4) and bulk RNA-seq (grey dots/bars). Error bars indicate s.e.m. from four independent RT-qPCR experiments. Rik*: *3632451O06Rik*

molecular determinants that may underlie the reported regional differences in VSMC growth factor response and disease susceptibility.

**Vascular region identity manifests in individual VSMCs.** The scRNA-seq dataset described above (Fig. 1) comprised 79 AA and 64 DT cells, allowing comparison of the transcriptional signatures associated with VSMCs from these regions at a single-cell level. Consistent with the region-specific gene signatures identified by bulk expression analysis, genes such as *Pde1c* and *Hand2* were almost exclusively detected in cells from the AA region and *Hoxa7* in cells from the DT region (Fig. 3a, top panel). However, other genes showing region-dependent expression levels in the bulk analysis were expressed at similar levels in a subset of cells from both regions, while the number of expressing cells was greater in either the DT (*Tnc*, *Calcrl*) or AA region (*Aspn*, *Gpc3*)

(Fig. 3a, lower panel). Therefore, the observed region-specific gene signatures could reflect the presence of different proportions of cells expressing these genes in each region. Alternatively, and not mutually exclusively with the above, each individual VSMC may bear an inherent regional signature.

To distinguish between these possibilities, we investigated whether VSMC regional identity can be reliably predicted from single-cell profiles. To this end, we employed random forest[19] (RF) analysis (Fig. 3b), a nonparametric tree-based machine learning method that is widely used for classification based on expression data, including scRNA-seq[20,21]. First, 25% of cells were randomly selected and set aside for the final testing of the prediction algorithm. Data for the remaining 108 cells was used to rank differentially expressed genes detected in the bulk RNA-seq analysis, based on their power to predict regional identity in individual cells. The influence of each of the top 30 predictive genes on RF classification accuracy is shown in Fig. 3c and

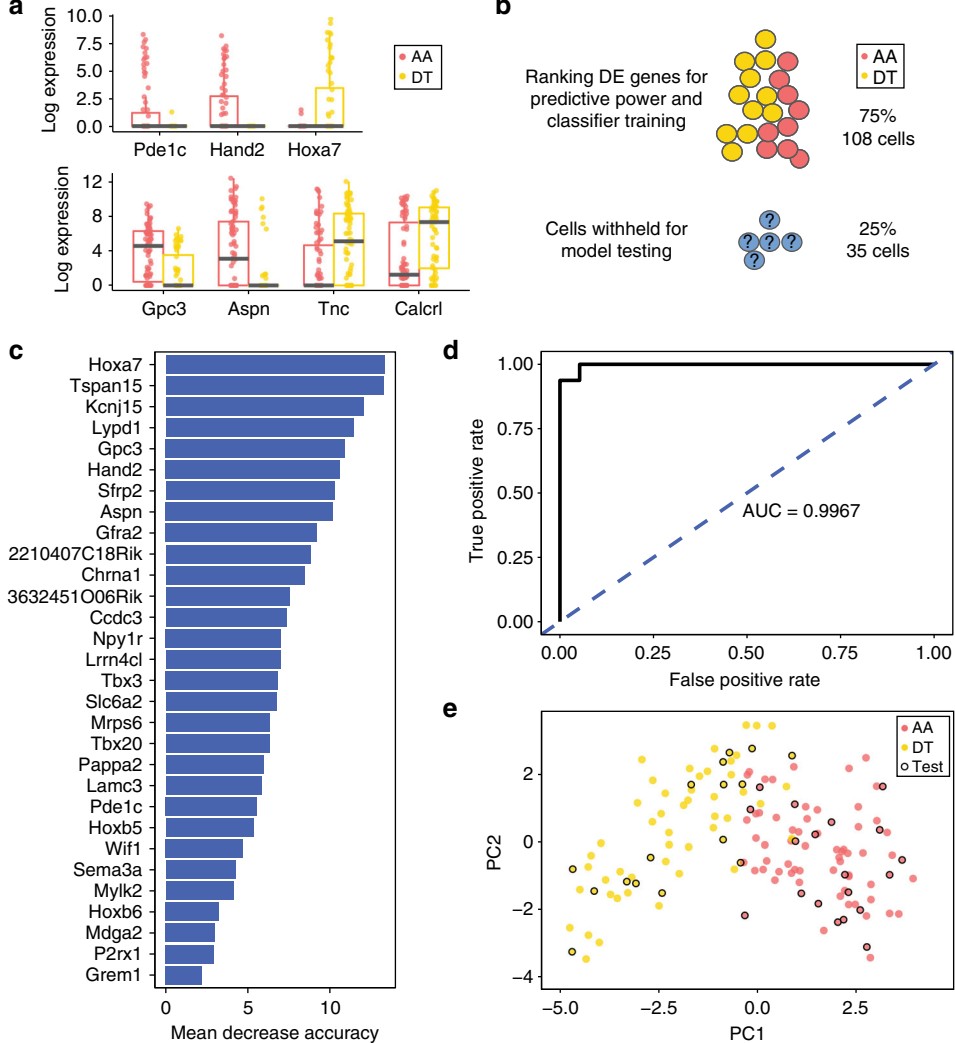

**Fig. 3** Single-VSMC transcriptomes reflect regional identity. **a** Boxplots showing log$_2$-transformed normalised expression of genes detected as differentially expressed between the aortic arch (AA, red) and descending thoracic aorta (DT, yellow) in bulk RNA-seq experiments across single VSMCs. Top panel: genes showing expression near-exclusively in cells from one tissue. Lower panel: genes that are expressed in different proportions of cells from the AA and DT. Median (centre line), first and third quartiles (bounds of box) and 1.5 interquartile range (whiskers) and individual data points (dots) are indicated. **b** Schematic of the random forest analysis. The regional identity of 75% of the cells was used in classifier training and refinement. The remaining 25% of cells (35 cells) were tested with the model showing the best classification performance with the 75% subset. **c** The out-of-bag mean decrease in classification accuracy for the 30 genes used in the final classifier. **d** ROC curve showing the performance of the classifier on the 25% subset of cells that were not used for model testing and refinement. **e** PCA plot based on the 30 genes used in the final classifier, showing AA (red) and DT (yellow) cells. Cells in the 25% test subset are circled in black

Supplementary Data 3a. These 30 genes were used in the final RF classifier; using additional genes for classification did not significantly increase classifier performance. The resulting RF model showed very high predictive power on the unseen 25% of the data, as demonstrated by the ROC curve (Fig. 3d), correctly classifying 17/18 AA cells and 14/15 DT cells set aside for model testing (Supplementary Data 3b). Consistent with this, a PCA plot based on the 30 RF classifier genes showed a clear segregation of AA from DT cells (Fig. 3e). The highly robust performance of the scRNA-seq-based classifier strongly indicates that features of regional identity are borne by individual cells, rather than reflect the differential composition of cell populations between the regions.

**VSMC heterogeneity within vascular regions.** We next examined whether cells within a specific vascular region showed

heterogeneity with respect to expression of other genes. To identify variably expressed genes robustly, we assessed the variance of log-transformed scRNA-seq counts for each gene across cells, relative to that expected from the estimated technical noise (modified from ref. [22]; Fig. 4a; see Methods for details). We identified 113 highly variably expressed genes (HVGs) in AA and 79 HVGs in DT VSMCs (adjusted $p$-value $< 0.05$; Supplementary Data 4ab). Notably, the HVG scores between the two regions were correlated ($r = 0.67$; Supplementary Fig. 4a) and only one of the identified HVGs (*Wif1*) was detected as differentially expressed between the two regions.

Many of the identified HVGs encode factors that have been directly linked to VSMC phenotype or have documented roles in cardiovascular disease, inflammation and VSMC biology (Fig. 4b and Supplementary Data 4c). For example, one of the top HVGs,

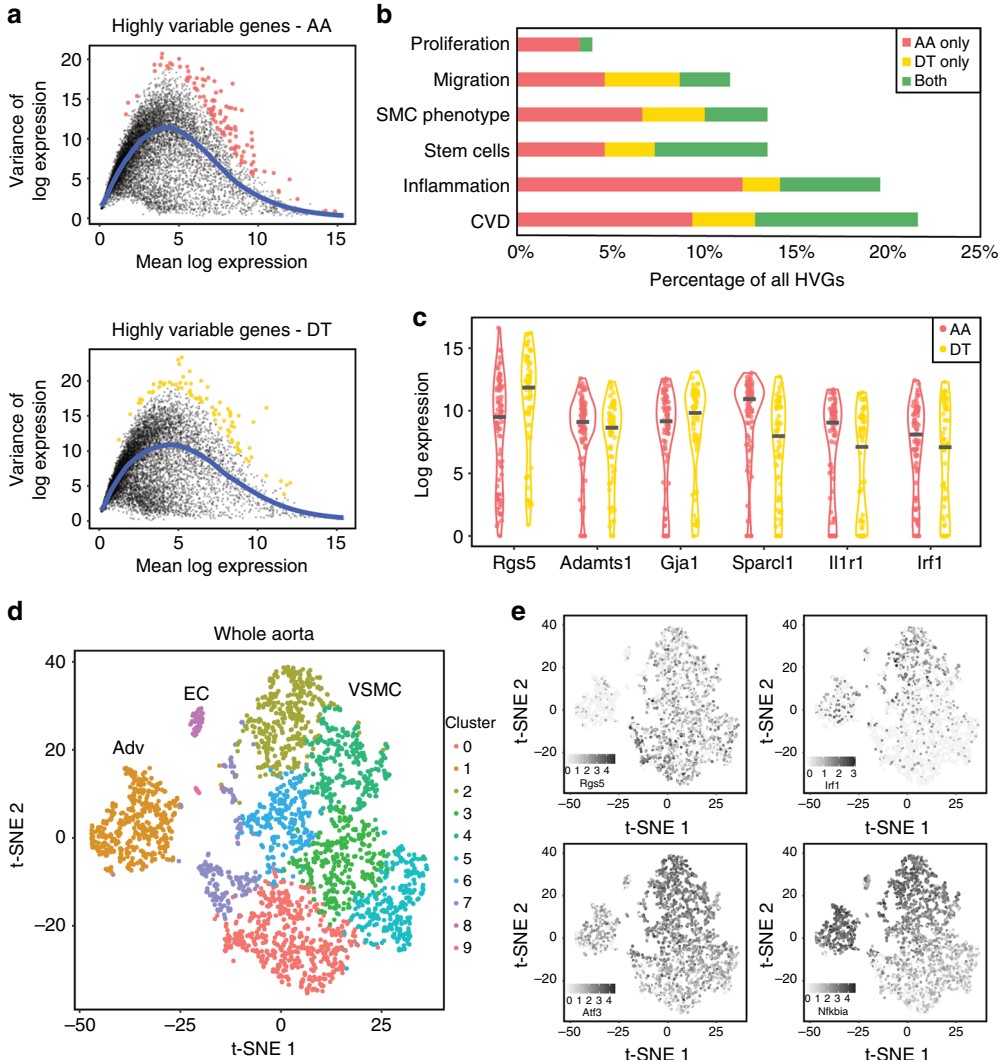

**Fig. 4** VSMCs show heterogeneous expression of genes implicated in cardiovascular disease. **a** Scatter plots showing the mean-variance relationship of log2-transformed normalised expression levels for each gene, with colour-highlighting of genes showing highly variable expression in the aortic arch (AA, red, top panel) and descending thoracic aorta (DT, yellow, lower panel). **b** Bar graph showing the implication of the identified highly variable genes (HVGs) in AA (red), DT (yellow) or both regions (green) in functions related to VSMC biology based on published literature (see Methods for details). **c** Dot plot showing log2-transformed normalised counts detected in individual VSMCs from AA (red) and DT (yellow) for selected genes that show variable expression across single cells. **d**, **e** t-SNE plot visualising a 10X Chromium dataset generated from 2846 unselected cells (gated as live using Zombie NIR staining and singlets using doublet discrimination) from the whole aortas of three tamoxifen-labelled *Myh11-CreERt2/Confetti* animals (pooled). **d** Clusters generated using graph-based clustering are colour-coded as indicated and adventitial (Adv), endothelial (EC) and VSMC (VSMC) populations are labelled. **e** Log-transformed expression levels of selected HVGs identified in AA and DT populations based on Fluidigm C1 data (*Rgs5, Irf1, Atf3, Nfkbia*), shown using a scale from light to dark grey

*Rgs5* (Fig. 4c), is induced by PDGF and known to control vaso-contractility by accelerating the inactivation of Gα-dependent signalling[23]. Interestingly, *Rgs5* is highly expressed in the descending aorta compared with the carotid arteries during development, but this regional difference is lost in adults[24]. Consistent with this, we found that *Rgs5* is expressed in subsets of single cells from both AA and DT (Fig. 4c). Other genes showing heterogeneous expression in at least one region include factors involved in adhesion and cell migration (such as *Adamts1*, *Gja1* and *Sparcl1*), as well as inflammation (such as *Il1r1* and *Irf1*) (Fig. 4c).

The robust HVG identification strategy employed above deliberately disfavours genes expressed at high levels in only a very small number of cells. However, such genes may be relevant to VSMC biology, given, for example, that only a small number of cells undergo clonal proliferation in response to injury or in atherosclerosis[8–10]. We therefore employed the "Distance to Median" (DM) method that ranks genes based on the squared coefficient of variation of gene expression counts given the mean[22,25] (Supplementary Data 5). This method ranked *Rgs5* (also identified using the robust approach) and *Vcam1* among the top HVGs in both AA and DT regions. *Vcam1* is expressed in endothelial cells, but has also previously been detected in VSMCs[26], where it is upregulated in response to pro-inflammatory cytokines[27]. Notably, another variably expressed gene highlighted in the DM analysis encodes Stem Cell Antigen 1 (*Ly6a/Sca1*; ranked within the top 1.2%).

We next assessed the expression of the identified heterogeneously expressed genes in a much larger dataset (~2800 cells) of dissociated aortic cells generated using 10X Chromium technology, which has a higher throughput, but reduced coverage compared with the Fluidigm C1-based analysis presented above[28]. As shown in Fig. 4d, t-SNE analysis of this dataset revealed clusters of arterial cells corresponding to VSMCs, endothelial and adventitial cell populations based on marker gene expression (including *Myh11/Cnn1*, *Cdh5* and *Pdgfra*, respectively, Supplementary Fig. 4b). *Myh11*-expressing VSMCs further segregated into seven clusters, highlighting the heterogeneity of this population. Remarkably, these clusters showed differential expression of 13% of the HVGs identified above, including *Rgs5*, *Irf1*, *Nfkbia* and *Adamts1* (22/147, Fisher's exact test $p = 2e-15$), and two genes prioritised using the DM method (*Rgs5* and *Plaur*) based on Fluidigm C1 data (Fig. 4e, see Methods for details).

Jointly, these analyses demonstrate that region-specific gene expression manifests at the level of individual VSMCs and may directly contribute to disease predisposition of the aortic arch. Additionally, many genes previously implicated in VSMC regulation and vascular disease development show cell-to-cell variation in expression levels in the vessel media. These highly variable genes could underlie the functional heterogeneity of VSMCs, such as that observed in atherosclerosis[8–10].

**A subset of VSMC-lineage cells express Stem Cell Antigen 1.**
The *Ly6a/Sca1* gene, which marks stem and progenitor cells in the vasculature[29–34] and other tissues[35], was identified as heterogeneously expressed in VSMCs with the DM method above (Supplementary Fig. 5a). Consistent with this, *Ly6a/Sca1* expression was detected in individual cells in the VSMC cluster in the 10X Chromium whole aorta analysis, in addition to the expected[30,36] abundant *Ly6a/Sca1* expression in the adventitial and endothelial clusters (Fig. 5a). Furthermore, 0.5–1% of medial cells ($n > 10$) stained positively with anti-Sca1 antibody in flow cytometry analysis (Supplementary Fig. 5b). To eliminate possible artefacts from antibody cross-reactivity, we also analysed transgenic Sca1-GFP animals, which express GFP from the *Ly6a/Sca1*

promoter[37]. Similar to the observations from antibody staining, 0.2–1.6% of medial cells from transgenic animals expressed GFP (Supplementary Fig. 5c, $n = 4$). To test whether GFP-expressing cells from the medial layer of Sca1-GFP animals have a VSMC identity, we sorted GFP-positive cells from either medial or adventitial samples and immunostained them for aSMA (a contractile VSMC marker). As expected, none of the 60 analysed GFP+ cells from the adventitial sample expressed aSMA. In contrast, 25–86% of medial GFP+ cells stained positive for aSMA (Fig. 5b, c). Taken together, these results suggest the existence of a rare subpopulation of Sca1-expressing VSMCs within the medial layer of healthy animals.

To further confirm the lineage affiliation of Sca1+ medial cells, we analysed Sca1 expression in the aortic media of mouse models that allow for genetic lineage labelling of VSMCs (*Myh11-CreERt2/EYFP*[38] and *Myh11-CreERt2/Confetti*[8]). These animals express the tamoxifen-inducible CreERt2 recombinase selectively in smooth muscle cells under control of the *Myh11* promoter. Activation of the Cre recombinase by tamoxifen administration therefore causes in VSMC-specific deletion of a stop cassette at either single-colour (EYFP) or multicolour reporters (Confetti), resulting in stable fluorescent protein expression (Fig. 6a). The specificity of the lineage label to the media was confirmed by analysis of 72 arterial sections from six Confetti-labelled animals (Fig. 6b, c), which detected only seven lineage-positive cells in the adventitial layer and one cell in the vascular endothelium. In contrast, recombination was efficient in medial cells (70–95% for the Confetti reporter and 40–90% for the EYFP reporter, Fig. 6d), consistent with previous data[8]. We confirmed by flow cytometry that a significant proportion of Sca1-expressing cells (S+) isolated from the media of *Myh11-CreERt2/EYFP* mice expressed the EYFP lineage label (Fig. 6e). We also detected a small number of *Ly6a/Sca1*+ cells in a 10X Chromium analysis of sorted *Myh11*-lineage labelled cells (Supplementary Fig. 6a), confirming the existence of a rare population of Sca1-positive VSMCs in the media.

**Single-cell expression profiles of Sca1+ VSMC-lineage cells.**
Due to the low frequency of *Ly6a/Sca1*+ VSMCs, the number of cells profiled using 10X Chromium technology was insufficient for robust analysis. We therefore index-sorted cells from tamoxifen-treated *Myh11-CreERt2/Confetti* animals based on Sca1 expression and profiled their transcriptomes using the Smart-seq2 protocol (Fig. 7a). To ensure that only single cells were analysed, we gated for cells expressing only a single colour of the lineage label (L) and confirmed singlet isolation by confocal microscopy (Supplementary Fig. 6b). In total, data for 92 Sca1-positive/lineage-positive cells (S+L+), as well as 27 Sca1-positive/lineage-negative (S+L-) and 36 "conventional" Sca1-negative/lineage-positive (S−L+) control cells, from six independent cell pools profiled in three experiments passed stringent quality control filters (Supplementary Fig. 6c-e, see Methods for details) and were used in further analysis.

*Ly6a/Sca1* transcripts were only detected in a subset of S+ cells (41/119, 34%, Fig. 7b), which could be due to either a lower sensitivity of scRNA-seq compared with FACS analysis or biological factors such as differences between transcript and protein stability. While all S−L+ cells expressed *Myh11* mRNA, as expected, S+L+ cells showed a broad range of *Myh11* transcript levels (Fig. 7b). *Myh11* was also expressed (more than 10 reads) in 6/27 of S+L- cells, suggesting that these cells represent the fraction of VSMCs that did not successfully recombine the lineage label after tamoxifen treatment.

Conventional VSMCs (S−L+; Fig. 7c, magenta triangles) and the majority of non-lineage-labelled Sca1-expressing cells (S+L-;

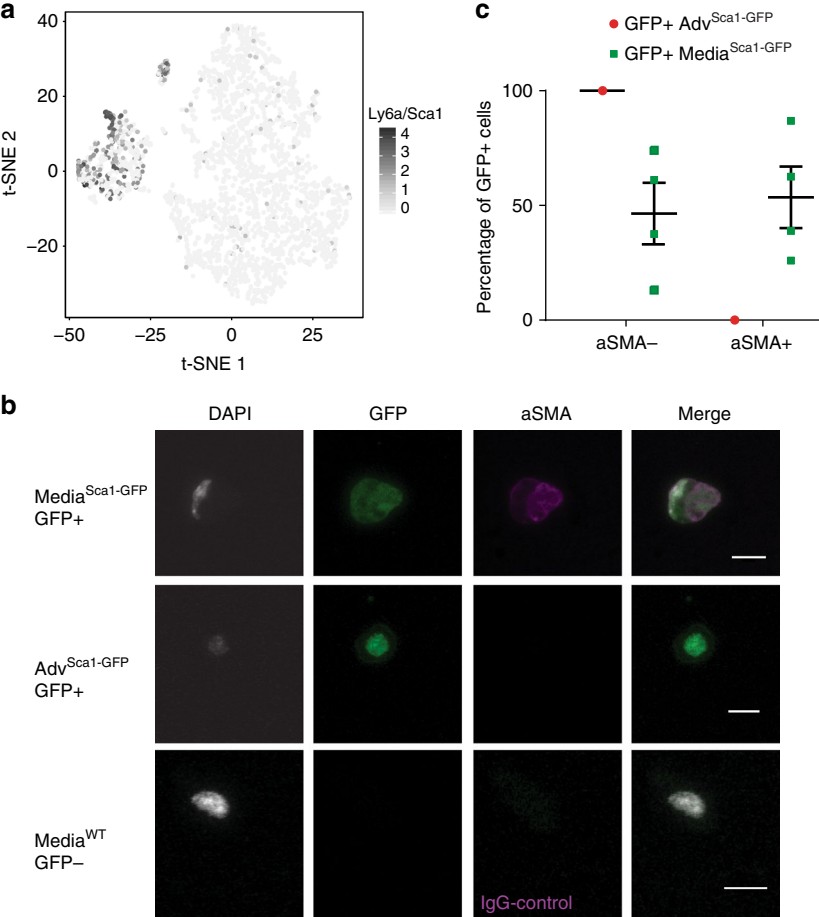

**Fig. 5** A subset of medial cells express Sca1. **a** t-SNE plot visualising the 10X Chromium dataset of unselected cells from whole aortas (shown in Fig. 4d), with log-transformed expression levels of *Ly6a/Sca1* in each cell colour-coded on a scale from light to dark grey. **b** Representative confocal images of GFP-positive cells isolated from the media (top row) or adventitia (Adv, middle row) of Sca1-GFP animals after immunostaining for aSMA. Signals for GFP (green), anti-aSMA (magenta) and nuclear DAPI (white) are shown individually and merged as indicated. The lower row shows GFP-negative medial cells isolated from wild-type animals and stained with isotype IgG as a control for staining and GFP detection. Scale bars are 10 μm. **c** Dot plot showing the percentage of aSMA– and aSMA+ cells in sorted GFP+ adventitial (Adv, red dot, n = 1) and medial cells (green dots, n = 4) from Sca1-GFP animals. Data for individual replicates and their mean values are indicated and error bars show s.e.m.

blue circles) formed distinct clusters in a PCA based on the levels of the 500 most variably expressed genes. In contrast, Sca1+ VSMC-lineage cells (S+L+; yellow squares) showed remarkable heterogeneity, with some S+L+ cells clustering together with conventional VSMCs, and others spreading away from them. These findings provided the initial evidence that VSMC-lineage Sca1+ cells are distinct from their non-VSMC-lineage counterparts and may show heterogeneity with respect to the contractile VSMC phenotype.

**Increased expression of VSMC response genes in Sca1+ VSMCs.** To identify gene signatures underlying the observed heterogeneity of S+L+ cells, we initially employed the robust HVG method used above for the analysis of the AA and DT VSMC populations. We identified 52 genes showing highly variable expression within the S+L+ population (Supplementary Fig. 7a,b and Supplementary Data 6), which fell into three network modules according to weighted gene co-expression network analysis (WGCNA, Fig. 7d and S7c, d). The module showing the highest level of co-expression consisted of 24 genes, including key contractile markers (*Myh11*, *Acta2*, *Tagln* and *Cnn1*; Fig. 7d), and we will refer to it as the "cVSMC network". As expected, the cVSMC network was expressed highly in conventional S−L+

VSMCs, variably across S+L+ cells and at low levels in S+L- cells (Supplementary Fig. 7e).

The reduced levels of cVSMC markers in a subpopulation of S+L+ cells that did not cluster with conventional S−L+ VSMCs (Fig. 7e) prompted us to search for genes that are upregulated as the contractile identity is lost. We assigned each cell a "cVSMC score" based on expression of the cVSMC network genes (Supplementary Data 7; see Methods for details). Using negative binomial regression, we detected 312 genes whose expression positively correlated with the cVSMC score in S+L+ cells (cVSMCpos), and 303 genes showing a negative correlation (cVSMCneg) (likelihood ratio test, fdr-adjusted $p$-value < 0.05; see Methods; Fig. 7f and Supplementary Data 8).

Genes that correlated positively with the cVSMC score were generally expressed at higher levels in S−L+ cells compared with S+L- cells and showed variable expression in S+L+ cells (Fig. 7g). As expected, cVSMCpos genes showed enrichment for ontology terms describing functional features of contractile VSMCs such as "muscle contraction" (*Tpm1, Mylk*), "actin filament organisation" (*Actn4, Dstn*) and "cell-substrate adhesion" (*Sdc4, Itgb1*) (Fig. 7h and Supplementary Data 9). In contrast, genes showing a negative correlation with the cVSMC score (cVSMCneg) were enriched for ontology terms associated with vascular development, potentially reflecting a less differentiated state compared with contractile

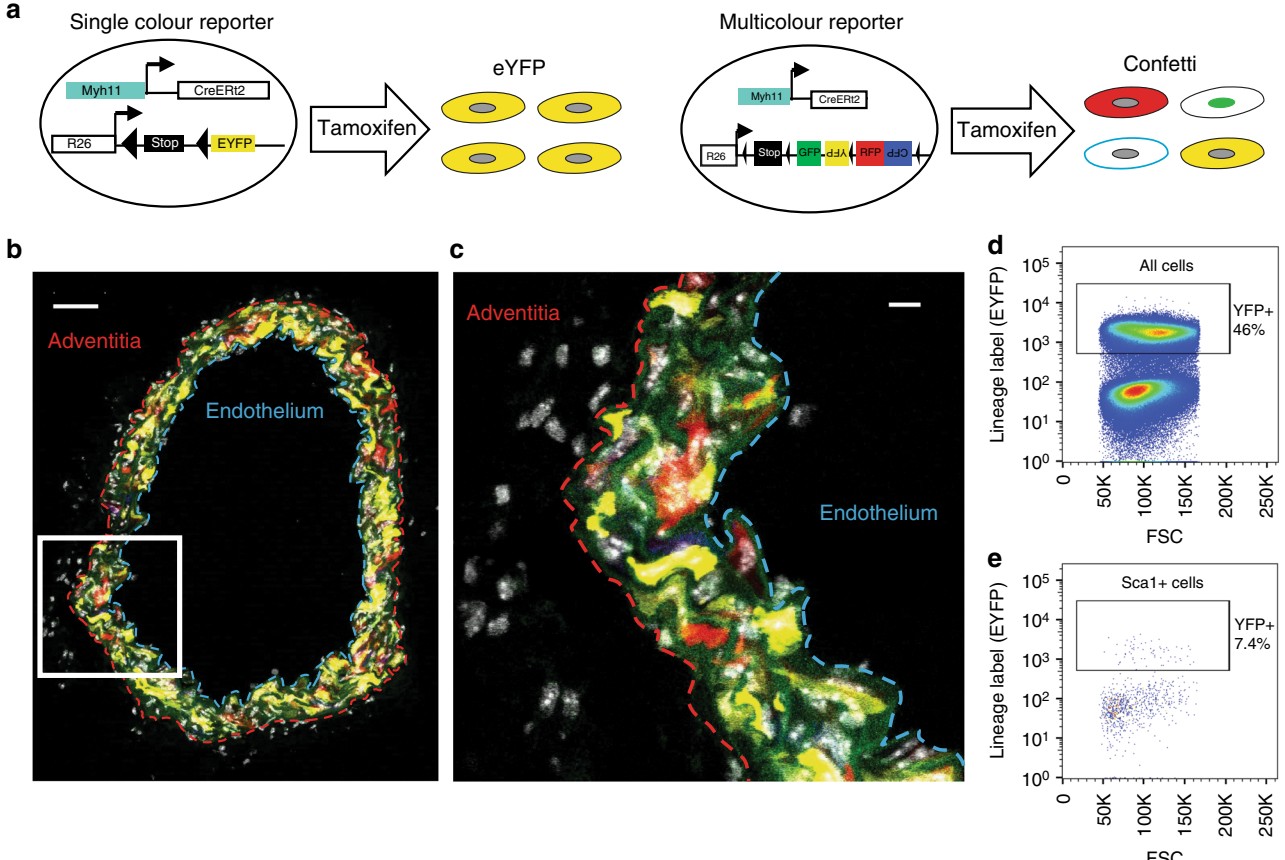

**Fig. 6** VSMC-lineage cells express Sca1. **a** Schematic showing strategies for lineage labelling of VSMCs using a Myh11-driven, tamoxifen-inducible Cre-recombinase (CreERt2). Tamoxifen treatment activates Cre recombinase activity, resulting in VSMC-specific excision of the stop codon in fluorescent reporter transgenes inserted into the Rosa26 locus (R26). Left panel, the single-colour EYFP reporter. Right panel, the multicolour Confetti reporter, which results in stochastic labelling of VSMC-lineage cells with one of four fluorescent proteins (GFP, YFP, RFP, CFP). **b**, **c** Maximum projection of a 12 μm transverse cryosection from the carotid artery of an *Myh11-CreERt2/Confetti* animal one week after tamoxifen labelling. Confetti fluorescent proteins are shown in red (RFP), yellow (YFP), blue (CFP) and green (nuclear GFP), elastic lamina autofluorescence in green and nuclear DAPI in white. The white boxed region in **b** is magnified in **c**, and the dashed lines show the medial-adventitial (red) and medial-endothelial (blue) borders in each panel. Scale bars are 50 μm (**b**) and 10 μm (**c**). **d**, **e** FACS plot showing forward scatter (FSC) and EYFP expression in all (**d**) or gated Sca1+ cells (**e**) isolated from the medial layer of aortas from *Myh11-CreERt2/Rosa26-EYF*P animals

VSMCs (Fig. 7h)[3,39]. Consistent with this, cVSMCneg genes included classical "synthetic" VSMC markers *Mgp*, *Col8a1* and *Spp1* and showed enrichment for terms associated with functional characteristics of phenotypically switched VSMCs, including migration (such as *Pak3, Igf1* and *Igfbp5*), proliferation ("cell growth", "wound healing") and secretion of extracellular matrix components ("vesicle organisation" and "extracellular matrix structure") (Fig. 7i and Supplementary Data 9). We also found enrichment for genes associated with activation of many signalling pathways (PI3K, small GTPases, Tgf-beta, chemokines), suggesting that the cVSMCneg gene signature reflects a responsive cell state.

Notably, a number of GO terms (including cell adhesion, migration and ERK1/2 signalling, Fig. 7h) showed enrichment among both cVSMCpos and cVSMCneg genes. However, we noted that in these cases the positively and negatively correlated genes promote different functions within the same broad pathways. For example, inspection of the focal adhesion pathway annotated in the KEGG database[40] suggested that cVSMCpos genes enhance stress fibre formation, while cVSMCneg genes promote cell proliferation (e.g., *Pak3*, Supplementary Fig. 8a). Similarly, in the MAPK pathway, genes that correlate positively with the cVSMC score are generally inhibitory, whereas

negatively correlated genes promote growth factor signalling (Supplementary Fig. 8b).

Collectively, our findings demonstrate that S+L+ cells are a heterogeneous cell population showing progressively lower levels of contractile VSMC genes (cVSMCpos) and higher expression of cVSMCneg genes that are associated with responses to diverse signals.

**Sca1 expression is a hallmark of VSMC stimulation**. To test whether Sca1 expression is linked to VSMC phenotypic modulation, we initially compared levels of *Ly6a/Sca1* transcripts in VSMCs isolated from healthy tissue with cultured VSMCs, representing a partial model of phenotypic switching. Increased *Ly6a/Sca1* expression in the cultured cells was evident by both RT-qPCR analysis (Fig. 8a) and in scRNA-seq datasets (Fig. 8b).

The increased Sca1 expression in cultured VSMCs could result from selective expansion of pre-existing Sca1+ medial cells or upregulation of *Ly6a/Sca1* transcription in contractile VSMCs upon phenotypic switching. To address this, we sorted Sca1 (GFP)-negative medial cells from Sca1-GFP animals and analysed GFP expression at different timepoints during in vitro culture. At day 3, no GFP expression was observed in the medial sample, while it was abundantly detected in sorted Sca1(GFP)-positive

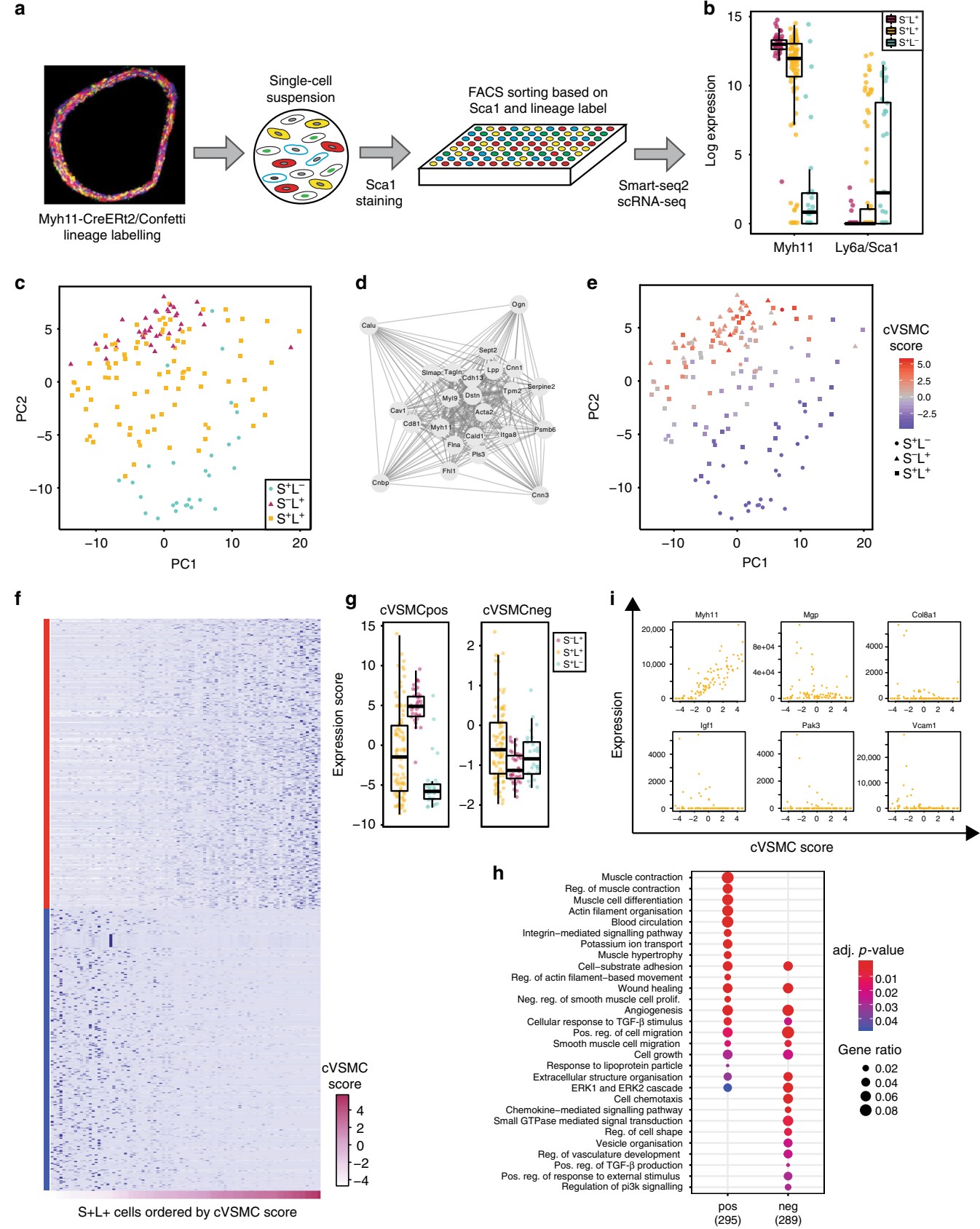

**Fig. 7** Transcriptional signature of S+L+ cells. **a** Schematic of the experimental strategy. A single-cell suspension of medial cells from *Myh11-CreERt2/Confetti* animals (a confocal image of an aortic cryosection is shown) was immunostained for Sca1 and index-sorted based on the expression of the lineage label and Sca1 to isolate individual cells for Smart-seq2 analysis. **b** Boxplots showing the log$_2$-transformed normalised levels of *Myh11* and *Ly6a/Sca1* transcripts detected in Sca1-negative, lineage label positive (S−L+, magenta), Sca1-positive, lineage label positive (S+L+, yellow) and Sca1-positive, lineage label negative cells (S+L−, blue). **c** PCA plot summarising the expression of the 500 most variable genes in S−L+, (magenta triangles), S+L+ (yellow squares) and S+L− (blue circles) cells. **d** The co-expressed cVSMC network module detected by weighted gene co-expression network analysis (WGCNA) based on 52 highly variable genes (HVGs) identified in S+L+ cells. Co-expression strength is indicated by edge thickness. **e** The PCA plot as in **c**, with cVSMC scores for S−L+ (triangles), S+L+ (squares) and S+L- (circles) cells colour-coded on a scale from blue to red. **f** Heatmap showing the expression of genes which positively (cVSMCpos, red side strip) or negatively (cVSMCneg, blue side strip) correlated with cVSMC score (likelihood ratio test, fdr-adjusted *p*-value < 0.05). Rows represent genes and columns represent cells ordered by their cVSMC score (colour-coded from light to dark purple, lower strip). **g** Boxplots showing the expression score of cVSMCpos (left) and cVSMCneg (right) genes for S−L+ (magenta), S+L+ (yellow) and S+L- (blue) cells. Expression scores are calculated based on PC1 as described in the Methods section. **h** Bubble plot of selected GO terms enriched in cVSMCpos (left) and cVSMCneg (right) genes. Dot size represents the number of genes overlapping with each GO term and the adjusted *p*-value is colour-coded from red to blue. The full list of enriched GO terms is given in Supplementary Data 9. **i** Dot plots showing the normalised expression of cVSMCpos gene *Myh11* and cVSMCneg genes *Mgp*, *Col8a1*, *Igf1*, *Pak3* and *Vcam1* across S+L+ cells, ordered by cVSMC score. Boxplots show median (centre line), first and third quartiles (bounds of box) and 1.5 interquartile range (whiskers) and data for individual cells (dots)

adventitial cells (Fig. 8c). However, longer exposure to culture conditions (7 days and more) induced GFP expression in medial cells, and on day 11, 15–28% were GFP-positive (Fig. 8c, d).

To test whether upregulation of *Ly6a/Sca1* is also observed in vivo, we analysed Sca1 expression by flow cytometry at different timepoints after induction of VSMC lineage labelling in *Myh11-CreERt2/EYFP* mice. We observed a small but significant increase in the fraction of VSMC-lineage cells that were Sca1-positive over time, as confirmed by logistic regression (Fig. 8e). The identified trend appeared stronger, rather than weaker, when the confounding effects of animal age were regressed out (Supplementary Fig. 9), suggesting that it was not due to age-related differences.

Finally, we tested whether Sca1 upregulation is also a feature of phenotypic modulation of VSMCs in vivo using a vascular injury model. Lineage-labelled *Myh11-CreERt2/EYFP* animals underwent carotid ligation injury, which results in the acute down-regulation of contractile genes and activation of proliferation[41]. Flow cytometric analysis of dispersed carotid arteries from these lineage-labelled animals demonstrated that Sca1 expression is induced in 10-45% of VSMCs (EYFP+) 8 days after ligation (Fig. 8f–h).

Taken together, these results provide in vitro and in vivo evidence that VSMCs upregulate Sca1 in response to stimuli such as culture conditions and vascular injury.

**Shared signature of healthy and plaque Sca1+ VSMCs.** The evidence for expanded numbers of Sca1+ medial cells following vascular injury (Fig. 8f–h) prompted us to investigate this population in atherosclerotic plaques. To this end, we used tamoxifen-labelled *Myh11-CreERt2/Confetti* mice on an *ApoE*$^{-/-}$ background, where atherosclerosis is induced by feeding a cholesterol-rich diet[8]. We isolated and digested atherosclerotic plaques and the underlying media from animals after 14-18 weeks of atherogenic diet and profiled the transcriptomes of sorted Confetti-positive VSMC-lineage cells from these samples with 10X Chromium technology. As shown in the t-SNE plot (Fig. 9a), most of the VSMC-derived plaque cells formed one large population, which could be further subdivided into nine clusters (clusters 0–8; Fig. 9a). Additionally, a clearly distinct population (cluster 9; Fig. 9a, pink) was formed by cells expressing macrophage markers (e.g., *Cd68*, *Lyz2* and *Fcer1g*).

In contrast to the low frequency of Sca1-expressing cells observed in healthy tissue, *Ly6a/Sca1* was detected in a significant proportion of VSMC-derived plaque cells (Fig. 9b). Most *Ly6a/Sca1+* cells were contained within cluster 7 (Fig. 9a, lilac), indicating that Sca1 expression marks a distinct type of VSMC-

derived plaque cells. We next examined whether Sca1-expressing VSMC-lineage cells in the plaques express the transcriptional signatures of their Sca1-positive counterparts (S+L+) found in healthy vessels. We found that cVSMCpos genes associated with the contractile phenotype were depleted in Sca1+ plaque cells compared with the majority of the profiled cells (Fig. 9c). In contrast, cVSMCneg genes (associated with response to signals) showed enrichment in the Sca1-expressing cluster (Fig. 9d). These results demonstrate that Sca1-positive VSMC-lineage cells from the plaques share transcriptional features of the rare S+L+ cells detected in healthy animals.

Notably, cluster 9 (Fig. 9a, pink) and a subpopulation of cells in the main cluster, which do not express *Ly6a/Sca1* (cluster 8; Fig. 9a, magenta), expressed cVSMCneg genes at higher levels than most plaque cells, suggesting that these cells could be related to the Sca1-positive population. Cluster 8 cells express chondrocytic genes (*Sox9*, *Ibsp*, *Chad*), consistent with a calcifying phenotype, whereas cells within cluster 9 have features of macrophages, as mentioned above. Notably, cVSMCpos genes were expressed at reduced levels in both of these cell populations compared with the S+L+ cell population (Fig. 9c, e), which may reflect their further progression away from the contractile state compared with the Sca1-positive cells.

Previous studies have suggested that the diversity of VSMC-lineage cells in the atherosclerotic plaque arises from clonal expansion of either one or very few cells[8–10]. However, it is not known whether Sca1-positive and Sca1-negative VSMC-derived plaque cells coexist in the same clones or, alternatively, arise from independent clonal origins. To test this directly, we stained plaques from *Myh11-CreERt2/Confetti/ApoE*$^{-/-}$ mice fed a cholesterol-rich diet for Sca1. We found that Sca1 is expressed in VSMC-derived cells within atherosclerotic lesions (Fig. 9f), consistent with previous observations[42]. Sca1-positive Confetti+ cells were located in the core region of the plaque, whereas Sca1 was not detected in the plaque cap (Supplementary Fig. 10). Yet, the clones that contained Sca1+ Confetti+ cells spanned both the cap and core regions, suggesting that Sca1-positive and Sca1-negative VSMC-derived plaque cells originate from a single cell. Taken together, these results indicate that dynamic modulation of Sca1 expression in VSMC-lineage cells is a hallmark of plaque development.

## Discussion

In this study, we have delineated VSMC heterogeneity and identified the molecular differences that may underlie their selective clonal expansion in disease[8–10]. We observe considerable heterogeneity between VSMCs within and between vascular beds

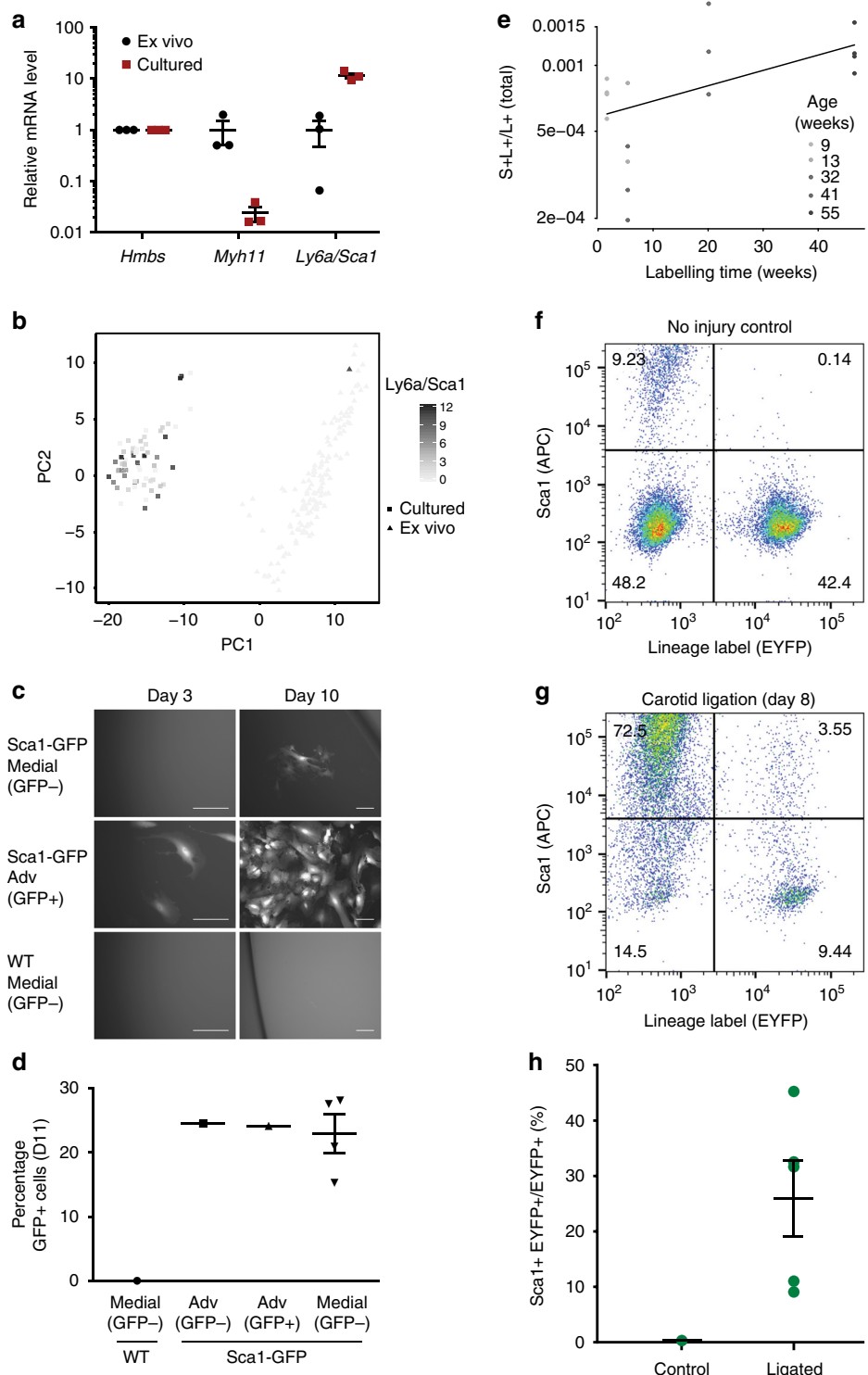

in healthy animals, and yet find that VSMCs uniformly show hallmarks of their vascular regional identity. We identify a rare population of Sca1-positive VSMC-lineage cells in healthy vessels and provide evidence from disease models that Sca1 expression marks VSMCs undergoing phenotypic switching.

The observed differential expression of transcription factors and signalling molecules between VSMCs from the aortic arch versus the descending thoracic aorta suggests distinct gene regulatory wiring between cells in these regions, which may underlie the reported regional differences in disease response[43]. This is consistent with classical transplantation experiments[44] and ex vivo studies[17] showing that gene expression and disease susceptibility of VSMCs is independent of vascular environment and growth conditions. In turn, the significant within-region VSMC heterogeneity in expression of genes involved in VSMC biology and disease, which we report, is consistent with that previously observed for G-protein coupled receptor expression by single-cell RT-qPCR[45]. Such heterogeneity is noteworthy as it indicates the existence of specific subsets of cells with particular disease-relevant properties. Sources of heterogeneity may include further

**Fig. 8** Sca1 is upregulated in response to VSMC stimulation in vitro and in vivo. **a** Relative expression (log$_2$-transformed) of *Myh11* and *Ly6a/Sca1* in ex vivo (black) and cultured mouse aortic VSMCs at passage 4–5 (red) determined by RT-qPCR, normalised to housekeeping gene expression (*Hmbs*). Lines show mean from analysis of three independent primary cultures, error bars show s.e.m. Differences in *Myh11* ($p = 0.001$) and *Ly6a/Sca1* ($p = 5.1e{-}10$) expression are statistically significant (student's *t*-test). **b** The PCA plot of single-cell expression profiles for ex vivo VSMCs (squares) and cultured VSMCs (triangles) shown in Fig. 1e, with expression level of *Ly6a/Sca1* colour-coded from light to dark grey. **c**, **d** Images (**c**) and GFP-signal quantification (**d**) of FACS-isolated medial cells from Sca1-GFP animals (sorted as GFP-negative, $n = 4$, top row in **c**) with Sca1-GFP adventitial (Adv, sorted as GFP– or GFP+ (middle row in **c**), tissue from four animals was pooled) and wildtype (WT) medial cell controls (sorted as GFP-, $n = 1$, lower row in **c**). Cells were cultured for 11 days before fixation and confocal imaging. **c** Epifluorescence images showing GFP signal after 3 or 10 days of culture. Scale bars are 100 μm. **d** Quantification of GFP signal in each population showing the number of GFP-positive cells as a percentage of the total number of DAPI-positive cells. Images for quantification were taken in a single *z* plane. Individual replicates and their mean are indicated and error bars show s.e.m. **e** Logistic regression analysis of the relationship between S+L+ cells and time after lineage labelling (logit-link logistic regression coefficient = 0.016+/−0.005 [mean+/−95% confidence interval], *p*-value based on Student's distribution = 2.56e−10). Trendline and data points, colour-coded by animal age (black gradient), are shown. Age was not included in the regression model presented here; the model accounting for both time after labelling and age is shown in Supplementary Fig. 9. **f**, **g** FACS plots showing EYFP and Sca1 (APC) expression in cells from the left common carotid artery (LCCA) isolated from tamoxifen-labelled *Myh11-CreERt2/EYFP* no injury controls (**f**) or eight days after ligation (**g**). **h** The percentage of lineage labelled cells (EYFP+) that expressed Sca1 in the LCCA isolated from ligated and no injury controls. Dots represents data from independent animals ($n = 5$ for each group), lines show means, and error bars s.e.m

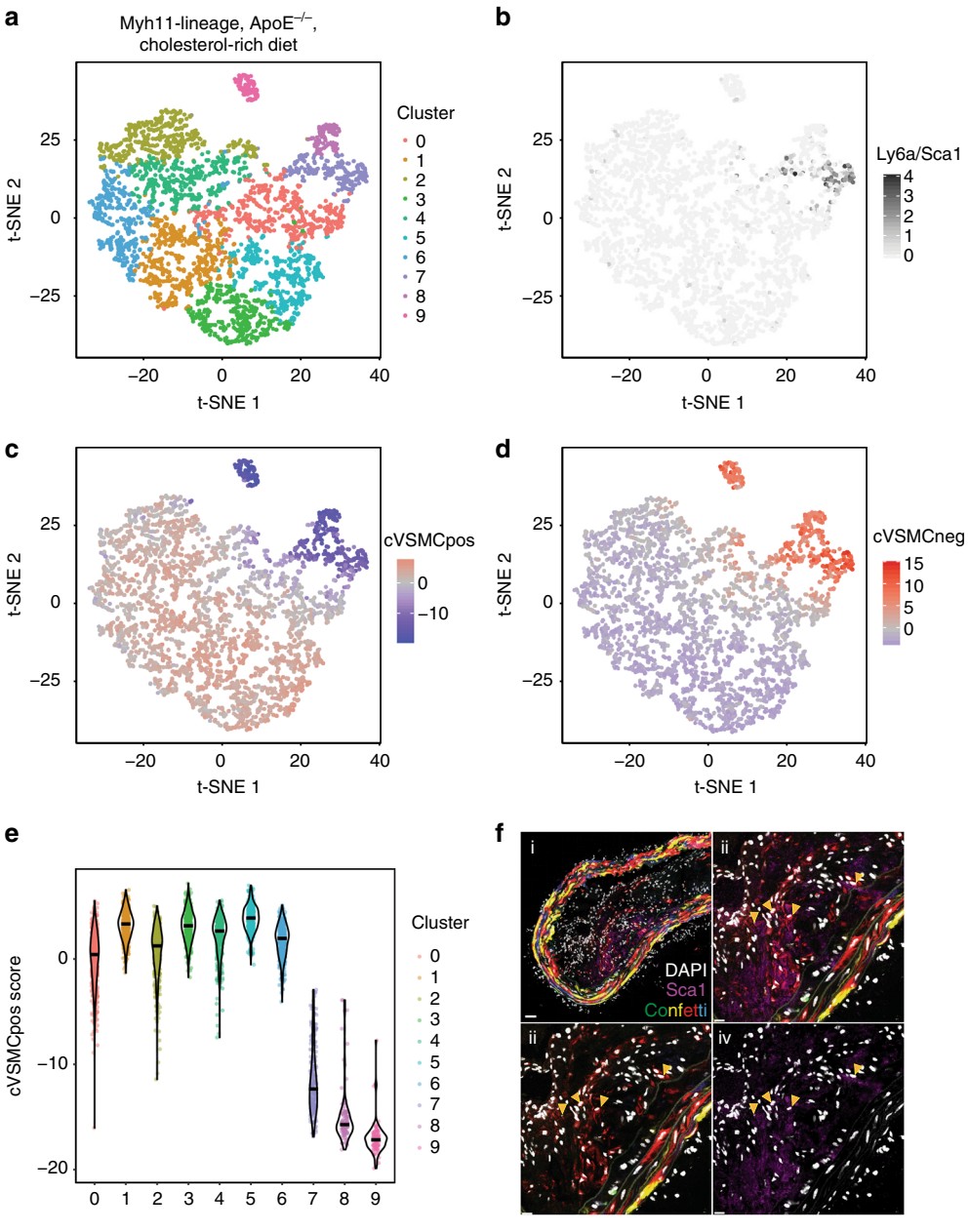

developmental differences within the VSMC lineage[46], differences in the cells' microenvironment and haemodynamic stress or stochastic factors. Analyses focused on individual genes are required to further investigate the mechanisms and functional consequences of VSMC heterogeneity in healthy vessels.

We explored the functional relevance of a rare subset of VSMC-lineage cells that express Sca1, which is encoded by one of the identified variably expressed genes (*Ly6a/Sca1*). We show that Sca1+ VSMC lineage cells constitute a heterogeneous cell population. Some of these cells exhibit a contractile VSMC (cVSMC) signature, as evidenced by expression of cVSMCpos genes. In contrast, other Sca1+ VSMC lineage cells express reduced levels of the contractile VSMC signature and instead manifest a "VSMC response signature" (represented by cVSMCneg genes), including markers of synthetic VSMCs. We further demonstrate that the transcriptional profile associated

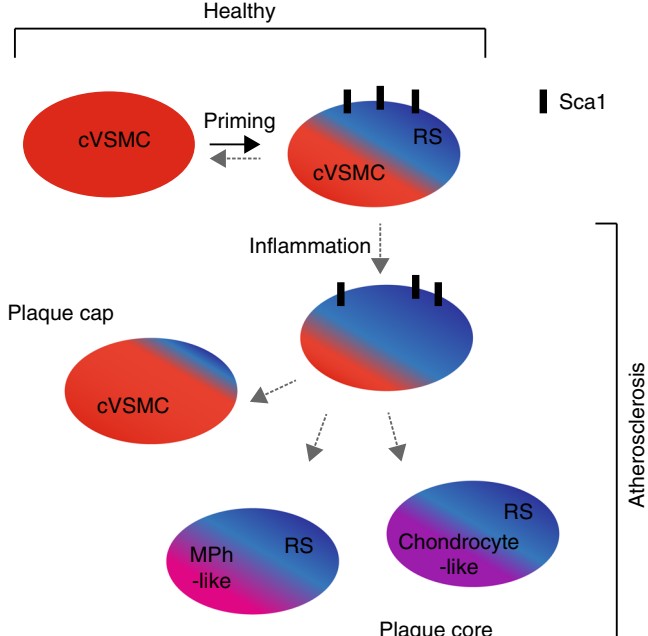

**Fig. 10** Model of VSMC priming and inflammation-induced phenotypic switching. Within the healthy vessel, priming of contractile VSMCs induces expression of Sca1 (black symbols) and Response Signature (RS, blue) genes, as well as the downregulation of the cVSMC signature (cVSMC, red). We propose that Sca1-expressing VSMC-lineage cells are hyper-responsive to inflammation and constitute an intermediate plastic population that potentially gives rise to phenotypically distinct, Sca1-negative VSMC-derived cells in the plaque (cap cells displaying the cVSMC phenotype, and core cells co-expressing the Response Signature with chondrocyte- (magenta) or macrophage-like genes (pink))

with these non-conventional Sca1+ VSMC-lineage cells in healthy vessels (VSMC response signature[High], cVSMC signature[Low]) is shared by Sca1+ cells within atherosclerotic plaques. We therefore propose that Sca1+ VSMC-lineage cells within healthy tissue could represent a more plastic state, which would readily respond to injury and inflammation (Fig. 10). Consistent with this idea, we show that Sca1 is upregulated in VSMCs upon exposure to stimuli known to induce phenotypic switching. In contrast, Sca1 is not expressed in VSMC-derived plaque cells that display the contractile signature (including cells in the cap). Sca1 is also absent in VSMC-derived plaque cells that have adopted alternative identities, including those expressing chondrocyte markers, which nonetheless also display the VSMC response signature characteristic of Sca1-positive cells. We therefore propose that Sca1 expression during plaque development could mark an intermediate VSMCs state, which might give rise to phenotypically distinct cells within lesions (Fig. 10).

Sca1-expressing cells in the vessel wall have been reported previously[34]. In particular, a medial "side population" isolated on the basis of reduced Hoechst staining (reflecting the expression of the ABCG2 transporter) was found to be enriched for Sca1-positive cells[32]. These "side population" cells expressed low levels of lineage-specific marker genes, but upregulated VSMC or endothelial markers in vitro in response to PDGF-BB/TGF-beta or VEGF, respectively, indicating an increased plasticity[32]. Expression of *Abcg2* was detectable in 6/92 of the S+L+ cells we analysed, suggesting some overlap between these cells and the "side population". S+L+ cells also showed increased expression of some endothelial-associated genes, including *Vcam1* and *Flt1*, but whether endothelial cell function can be induced in these cells remains to be tested. Vascular wall-derived Sca1-expressing cells capable of differentiating into SMCs in vitro and in vivo have also been detected in the adventitia (AdvSca1)[29–31]. However, transcriptional profiling of a subset of the AdvSca1 cells (marked by Sonic Hedgehog (Shh)-response gene *Gli1*) showed expression of genes that are either not expressed in S+L+ cells (e.g., *Eng/Cd105* and *Sox2*), or that form part of the cVSMC signature, which is expressed in both S+L+ cells and contractile VSMCs (*Smtn*, *Tagln*, *Cnn1*, *Itgb1*)[31]. Finally, Sca1 is expressed by a population of angiogenic progenitors residing in the vascular wall[34]. The markers of these cells (*Pecam1/Cd31*, *Kit/cKit*, *Cdh5/VE-Cad-herin*, *Eng/Cd105*)[34] do not form part of the VSMC response signature identified in S+L+ cells, but notably, were detected in many S+L- cells. In conclusion, the transcriptional signature of S+L+ cells suggests that medial Sca1-expressing cells are distinct from adventitial and angiogenic progenitor cells. However, further analyses are required to comprehensively compare the previously reported cell populations in the vessel wall with the Sca1+ VSMC-lineage cells detected in our study.

A number of independent lineage tracing experiments have demonstrated that cells expressing contractile VSMC markers

**Fig. 9** The transcriptional signature of S+L+ cells from healthy vessels is expressed in VSMC-derived plaque cells. **a** t-SNE visualisation of 10X Chromium dataset (3314 cells) of lineage-positive cells isolated from aortic plaques and the underlying media in *Myh11-CreERt2/Confetti* mice that were first tamoxifen-treated and then fed a cholesterol-rich diet for 14 or 18 weeks (tissue from three and two animals respectively was pooled for the analysis). The cells are grouped into a large population composed of 9 clusters, including one enriched for chondrocytic genes (cluster 8, magenta), and a smaller, distinct population, which was enriched for macrophage genes (cluster 9, pink). **b** t-SNE plot of 10X Chromium dataset from **a** with *Ly6a/Sca1* expression colour-coded from light to dark grey. **c, d** t-SNE plot of 10X Chromium from **a**, with expression scores for cVSMCpos (**c**) or cVSMCneg (**d**) genes colour-coded on a blue to red gradient. **e** Violin plots showing the expression score of cVSMCpos genes in cells mapping to each cluster shown in **a**. **f** Confocal image of a cryosection of the descending thoracic aorta from a *Myh11-CreERt2/Confetti/ApoE^-/-* mouse fed a cholesterol-rich diet for 30 weeks and immunostained for Sca1. Signals for fluorescent Confetti proteins (GFP, green; RFP, red; YFP, yellow and CFP, blue) are shown in i–iii, nuclear DAPI (white) is shown in all panels, and anti-Sca1 (magenta) is shown in i, ii, and iv. The region outlined in i is magnified in ii–iv, with arrows pointing to cells that are double-positive for the Confetti lineage label (RFP) and Sca1. Image in i is a maximum projection of 16 z-slices (2 μm each) and ii–iv show a single 2 μm Z-slice. Scale bars are 40 μm (i) or 15 μm (ii–iv)

(*Myh11*, *Tagln*) generate the majority of cells within atherosclerotic lesions[8–10,38,42,47–49]. However, other studies suggest that specialised Sca1+ progenitor populations also give rise to vascular lesions[30,31,50], possibly alongside VSMCs in a context-dependent manner[51,52]. Our observation that a small proportion of Myh11-lineage labelled VSMCs in healthy vessels express Sca1, and that Sca1 is upregulated in VSMCs in response to stimulation, may provide an explanation for these apparently discrepant observations. The co-expression of Sca1 and the VSMC response signature with the Myh11-lineage label both in healthy vessels and within plaques suggest that observed Sca1+ cells in vascular disease models are likely to originate from phenotypically modulated VSMCs. However, development of specific, dual-lineage tracing models is required to examine the functional role of the previously reported progenitor populations and the S+L+ cells we have identified here in order to resolve this definitively.

The hallmarks of VSMC heterogeneity revealed in our study enable the isolation and functional analysis of specific VSMC subpopulations, paving the way for selective targeting of causative cell populations in vascular disease. In particular, the identified VSMC response signature may provide insight into the function and regulation of disease priming. Our study also highlights candidate genes for functional testing of how VSMC heterogeneity affects the response to inflammatory signals in disease and can be used to identify corresponding plastic cell populations in humans for targeting and diagnosis. Beyond the immediate findings of our study, the single-cell transcriptomes generated here for the whole aorta and VSMC-lineage cells in healthy vessels and atherosclerotic plaques will enable further explorations of vascular cell heterogeneity and function.

## Methods

**Animals and tissue dissection**. All experiments were carried out according to the UK Home Office regulations under project licence PPL 70/7565 and have been approved by the University of Cambridge Ethical Reviews Committee. C57Bl/6 animals were purchased from Charles River. *Myh11-CreERt2*, *Rosa26-Confetti* (Confetti), *Rosa26-EYFP* (EYFP), *ApoE^-/-*, *Sca1-GFP* animals have been described previously[8,37,38]. Males (the *Myh11-CreERt2* transgene is Y-linked) received 10 intraperitoneal injections of 1 mg/ml tamoxifen in corn oil, typically at 6–8 weeks of age, if not indicated otherwise, for lineage labelling. Aortas from young healthy male mice (8–14 weeks), unless otherwise indicated, were dissected free of fat and connective tissue. The aortic arch (AA, the aortic segment from just left of the branchpoint for the brachiocephalic artery to just right of the branch point for the left subclavian artery) and descending thoracic (DT, the straight aortal segment from after the arch has clearly ended until the diaphragm) aortic segments were isolated and processed separately in experiments comparing vascular regions. The animals used for analysis of S+L+ cells over time in vivo, were 9–55 weeks of age and were sacrificed 1.6–46.6 weeks post-tamoxifen labelling. For analysis of plaque cells, *Myh11-CreERt2/Confetti/ApoE^-/-* animals were treated with tamoxifen as described above and fed a cholesterol rich diet (Special Diets Services, containing 21% fat and 0.2% cholesterol) for 14–18 weeks, starting one week after the last tamoxifen injection.

**Tissue processing, cell staining and flow cytometry**. To generate single-cell suspensions, endothelial cells were removed manually from dissected aortas using a cotton bud and vessels incubated for 10 min in Collagenase Type IV (1 mg/ml, Life Technologies) and porcine pancreatic elastase (1 U/ml, Worthington) in DMEM to allow for separation of the adventitia and medial cell layers. Tissue isolated from 1 to 10 animals was pooled and further digested for 1–2 h to achieve a single-cell suspension, which was filtered through a 40 μM cell strainer. Single-cell suspensions of medial and adventitial cells were incubated with FcX (1:100, Biolegend) to block FcRIII binding before staining with APC-conjugated anti-Sca1 (1:10, Miltenyi 130-120-343) or isotype control antibody (1:10, Miltenyi 130-102-655). For Smart-seq2 experiments, samples were pre-stained with Zombie-NIR (1:100, Biolegend) to eliminate dead cells. Stained cells were analysed using Fortessa (*Myh11-CreERt2/Confetti*) or C6 (wild-type and *Myh11-CreERt2/Rosa26-EYFP*) flow cytometry analysers or index-sorted on an Aria-fusion flow cytometry-assisted cell sorter (BD Bioscience). The gating strategy is outlined in Supplementary Fig. 11. Isolation of single cells was verified by sorting *Myh11-CreERt2/Confetti* cells directly into a poly-L-Lysine coated chamber slide followed by cell fixation, DAPI staining and analysis by confocal microscopy. GFP-positive cells from enzyme dispersed single-cell suspensions of medial cells from Sca1-GFP transgenic animals were sorted directly into poly-L-Lysine coated chamber slides and fixed in 4%

paraformaldehyde for 20 min. Cells were washed in PBS and incubated for 1 h with blocking solution (10% goat serum in 0.1% BSA in PBS) then incubated overnight at 4 °C with biotin-conjugated anti-aSMA (1:200, Abcam ab125057) in blocking solution. Cells were then washed in PBS and incubated with streptavidin-conjugated streptavidin-conjugated Alexa Fluor-647 (1:200, Biolegend 405237 for one hour, before DAPI nuclear staining and mounting in ProLong Diamond antifade mountant (Thermo Fisher). In parallel, GFP-negative cells from Sca1-GFP animals were sorted into 96-well plates and cultured in DMEM with 10% FCS at 37 °C/5% $CO_2$. The cells were observed daily and imaged using an inverted fluorescence microscope at day 0, day 3, day 6 and day 10. After 11 days of culture, the cells were fixed, DAPI stained and mounted as described above.

**Analysis of Sca1 expression after carotid ligation surgery**. Tamoxifen-injected *Myh11-CreERt2/Rosa26-EYFP* animals were subjected to carotid ligation surgery[8,41]. Animals were anaesthetised using 2.5–3% isofluorane (by inhalation) and given a pre-operative analgesic (Temgesic). The left common carotid artery was tied firmly with one knot using 6-0 silk suture just below the bifurcation point and animals sacrificed 8 days after surgery. The carotid arteries were removed, dissected free from adipose and connective tissue and whole arteries were digested as described above until a single-cell suspension was reached. The dissociated cells were washed once in FACS buffer (1% BSA in PBS), incubated with FcX (1:100, Biolegend) to block FcRIII binding, stained with APC-conjugated anti-Sca1 (1:10, Miltenyi 130-120-343) or isotype control antibody (1:10, Miltenyi 130-102-655) and filtered through a 40 μm cell strainer. Stained cells were analysed on an Aria-fusion (BD Bioscience) and FlowJo V10 software was used for data analysis and quantification.

**Analysis of aortic cryosections**. After tissue dissection, aortas were fixed in 4% paraformaldehyde in PBS for 20 min, followed by overnight cryoprotection in 30% sucrose in PBS, 1 h incubation in a solution of 50% TissueTek O.C.T in 30% sucrose in PBS, 1 h incubation in 100% O.C.T and then snap-frozen in O.C.T. Serial transverse cryosections (12 μm) were then cut onto Superfrost Plus microscope slides, washed in PBS, stained with DAPI at 1 μg/ml in PBS and mounted in RapiClear 1.52 followed by confocal imaging. Cryosections from the aorta of animals fed a cholesterol-rich diet were stained for Sca1 before mounting: cryosections were washed in PBS, blocked for 1 h (10% goat serum in 0.1% BSA in PBS), incubated overnight at 4 °C with anti-Sca1 (1:200, Biolegend 108101) or Alexa Fluor 647-coupled anti-rat IgG2a (1:200, Biolegend 400526) in blocking solution, washed in PBS and incubated with Alexa Fluor 647-coupled anti-rat antibody (1:1000, Abcam ab150167) for one hour, washed in PBS, incubated with DAPI (1 μg/ml in PBS) to counterstain nuclei and mounted in RapiClear 1.52. For quantification of Confetti signal outside of the medial layer (defined as the area between the internal and external elastic lamina), four cryosections were analysed from each of two different parts of three vascular regions (AA, DT and carotid arteries) of six *Myh11-CreERt2/Confetti* animals either 1 or 16 weeks after tamoxifen labelling (three animals per time point, 72 sections in total). No difference in recombination frequency or detection of Confetti-positive cells in the endothelial and adventitial layers were detected between the two chase timings.

**Imaging and image analysis**. Live cultured cells were imaged using an Olympus IX71 fluorescence microscope with HCImage Live software (Fig. 8c). Confocal imaging was done using a Leica SP8 with LASX software and a 40x objective (Figs. 5b, 6b, c, 7a, 9f, Supplementary Fig. 6b and Supplementary Fig. 10) or Zeiss 700 with Zeiss Zen software and a 20x objective (images used for quantification in Fig. 8d). Cell counting and image processing was performed using ImageJ or Imaris software. Staining intensities in Fig. 5c were calculated from maximum projections of 14-24 1 μm z-stacks, representative images are shown in Fig. 5b.

**Bulk gene expression analysis**. Dissected aortas were immediately transferred to RNAlater followed by isolation of AA and DT segments before manual removal of the adventitial and endothelial cell layers. The cleaned medial layer from 3 to 5 animals was then lysed in Trizol (Thermo-Fisher) and RNA isolated. For bulk RNA-seq, the extracted RNA was cleaned on a RNeasy column (Qiagen) and quality-assessed on a Bioanalyzer (Agilent, RNA integrity number [RIN] 7.8-9). Sequencing libraries were made from 550 ng total RNA using the TruSeq Stranded mRNA Library Prep Kit (Illumina) and sequenced using MiSeq (Illumina). In vitro cultured samples were isolated from enzymatically dispersed VSMCs that had been cultured for 4–5 passages in DMEM supplemented with 10% foetal calf serum, glutamine and penicillin. For RT-qPCR analysis, cDNA was synthesised from 0.5 μg RNA using the Quantitect® RT kit (Qiagen) according to the manufacturer's instructions. qPCR was performed using SYBR Green PCR master mix (Applied Biosystems, Life Technologies) and a Rotor-Gene cycler (Qiagen) in duplicate reactions. Primer sequences are provided in Supplementary Table 1. Quantification was performed relative to a standard curve generated from serial dilutions of ex vivo whole-aorta cDNA and normalised using the weighted average of housekeeping genes *Hprt1* and *Yhwaz* or *Hmbs*.

**Analysis of bulk RNA-seq data**. Reads were aligned to the GRCm38 mouse genome using Tophat[53] v2.1 and read alignments per gene were counted using

Seqmonk v1.42 (http://www.bioinformatics.babraham.ac.uk/projects/seqmonk). Differential gene expression analysis was performed with Bioconductor R package DESeq2[54] v1.12. Genes with log₂ fold-change > 1 and fdr-adjusted $p$-value < 0.01 were identified as differentially expressed.

**Single-cell RNA-sequencing.** Fluidigm C1: Single-cell suspensions of medial AA and DT samples from wild type C57BL/6 males (5–7 per experiment) were prepared as described above and processed using a Fluidigm C1 system. Cell suspensions (100 cells/µl) were loaded onto medium-sized (17–25 µm) Auto Prep Arrays (Fluidigm) and processed according to the manufacturer's instructions. The loaded arrays were visually assessed under an inverted microscope to select capture sites containing a single cell, yielding a 40–70% capture efficiency. Cells included in the analysis were from two independent experiments for each aortic region. Cells were processed using the SMARTer® Ultra™ Low RNA Kit (Clontech) and amplified cDNA was isolated from the arrays.

Smart-seq2: For the analysis of the Sca1+ medial population isolated from *Myh11-CreERt2/Confetti* animals, single-cell suspensions of the medial layer from 5 to 7 animals per experiment were prepared and stained as described above. Individual Sca1+ and control Sca1-negative cells expressing a single Confetti-lineage label (L+) or Sca1+ cells expressing no Confetti marker (L−) were FACS-sorted on an Aria-Fusion flow cytometer (BD Bioscience) into separate wells of a 96-well plate. Sca1+ and Sca1− cells were included on the same plate and processed together to generate amplified cDNA using the Smart-seq2 protocol carried out as described previously[55], with the following minor modifications: Primescript (Clontech) was used for reverse transcription, 24 PCR cycles were used for amplification of cDNA, and ERCC control RNA (Invitrogen) was added (1:40,000,000 or 1:80,000,000 dilution in RT-mix). The analysed cells were from three independent experiments using six different medial cell suspensions.

10 Chromium: For analysis of whole aortas, aortas from three tamoxifen-labelled *Myh11-CreERt2/Confetti* males were dissected free of fat and connective tissue and subjected to enzymatic digestion. Single-cell suspensions were pooled, stained with Zombie-NIR (1:100, Biolegend), filtered through a 40 um filter and 20,000 singlet, live cells isolated by FACS-sorting on an Aria-Fusion flow cytometer (BD Bioscience). Note that the Confetti lineage label was not used for sorting. For analysis of VSMC-lineage labelled cells from healthy vessels, medial single-cell suspensions from three tamoxifen-labelled *Myh11-CreERt2/Confetti* males were generated as described above and FACS sorted on an Aria-Fusion flow cytometer (BD Bioscience) for expression of live (Zombie-negative), singlets expressing a single Confetti colour only (gating strategy provided in Supplementary Fig. 11). For 10X Chromium analysis of VSMC-derived atherosclerotic plaque cells, plaques were manually isolated by dissection from *Myh11-CreERt2/Confetti/ApoE⁻/⁻* animals that had been tamoxifen labelled and then fed a cholesterol-rich diet for 14 or 18 weeks (as described in Chappell et al.[8]). Plaques were digested as described[56] and plaque cells from two-three animals per time point were pooled and processed in parallel, filtered through a 40 µm filter and 20,000 singlet cells expressing a single Confetti protein isolated by FACS-sorting on an Aria-Fusion flow cytometer (BD Bioscience) (gating strategy provided in Supplementary Fig. 11, no selection for Sca1-expression included). Sorted cells were pelleted and resuspended in 35 µl PBS with 0.05% BSA and loaded onto a 10X Chromium system.

After quality assessment by Bioanalyzer (Agilent) and Quant-iT PicoGreen (Thermo Fisher) quantification, sequencing libraries were prepared from amplified cDNA from Fluidigm and Smart-seq2 protocols (approximately 6 ng) using the Nextera library prep kit (Illumina). Libraries were analysed by paired-end sequencing on an HiSeq 2500 system (Illumina). Sequencing libraries were directly generated by the 10X Chromium system and subsequently sequenced on a HiSeq 4000 system.

**QC and normalisation of Fluidigm C1 and Smart-seq2 data.** Reads were aligned to the GRCm38 mouse genome using *Tophat*[53] v2.1 and counted using htseq-count[57] v0.8. Single-cell transcriptomes generated with the C1 Fluidigm platform were quality-controlled based on the total read count (more than 1 million and fewer than 3.5 million read pairs), the number of genes detected per cell (above 5000 and below 9500), the percentage of reads mapping to genes (more than 80%) and exons (more than 50%), as well as the percentage of mitochondrial reads (less than 20%). Single-cell transcriptomes generated using the Smart-seq2 protocol were quality-controlled based on the total number of reads (more than 100,000) and genes (more than 1500) detected and the percentage of reads mapping to ERCC controls (below 30%). Gene-level read counts were normalised using the computeSumFactors function from the Bioconductor R package *scran*[22] (v1.2 for Fluidigm C1 data and v1.8 for Smart-seq2 data). For joint PCA analysis of single-cell and control samples (tube controls), estimateSizeFactorsForMatrix function from DESeq2 v1.14 (locfunc = shorth) was used for normalisation.

**Random forest analysis.** Cells were split into training (108 cells; 75%) and test (35 cells; 25%) sets. Recursive feature elimination with 10-fold cross-validation was performed on the training set using the caret R package (v6.0). This procedure identified the 30 most relevant genes, which were then used as features for training a random forest classifier (number of trees = 1000) with 10 times repeated 10-fold

cross-validation using the randomForest R package v4.6. The R package ROCR v1.0 was used to generate the ROC plot (Fig. 3d).

**Detection of highly variable genes (HVGs).** HVG analysis was performed separately on AA, DT and S+L+ cells. We used a strategy based on variance decomposition of log-transformed normalised expression counts into technical variance (learned from data as a function of log-transformed mean count for the AA and DT cells and estimated using ERCC spike-in controls for S+L+ cells) and the biological variance of interest, whereby HVGs are defined as genes with the biological component of variance significantly greater than zero. The analysis was performed using the trendVar and decomposeVar functions in the Bioconductor R package scran[22] (v1.2 for Fluidigm C1 data and v1.8 for Smart-seq2 data). To obtain robust estimates, we repeated the analysis 1000 times using a 90% subset of cells for each run. The $p$-values from the one-tailed HVG tests for each run were non-independent and so could not be combined using standard methods. We therefore adopted an approach based on work by Licht and Rubin[49], whereby the $p$-values for each gene $g$ and run $i$ are transformed into z-scores $z_{g,i}$ using the quantile function of the standard normal distribution. Gene-level z-scores are then computed as $z_g = \frac{\overline{z_{g,i}}}{T_{g,i}}$, where $\overline{z_{g,i}}$ is the mean z-score and $T_{g,i}$ is the total variance of z-scores, respectively, across $m = 1000$ runs for a given gene. The total variance $T_{g,i}$ is taken to be:

$$T_{g,i} = \overline{\mathrm{Var}(Z_i)} + \mathrm{Var}(Z_{g,i}), \tag{1}$$

and estimated as:

$$T_{g,i} = 1 + \frac{1}{m-1}\sum_i^m (\overline{z_{g,i}} - z_{g,i})^2, \tag{2}$$

given that $Z_i \sim N(0,1)$. The corresponding gene-level $p$-values are then obtained from the probability density function of the standard normal distribution and adjusted for multiple testing using the Benjamini-Hochberg procedure. Genes with adjusted $p$-values below 0.05 were considered HVGs. Notably, this approach produced very similar results to Hartung's method for combining dependent $p$-values[58] (as implemented in R package punitroots[59] v0.0-2), but yielded more conservative estimates for genes showing a large variance of HVG test p-values observed across runs.

We also ranked genes based on the difference between the squared coefficient of variation (SCV) of their expression values and the median SCV expected given their sample mean, using an approach proposed by Kolodziejczyk et al[25]. and implemented in the function DM in scran[22] v1.6.

**Annotation of HVGs.** The functional annotation of the 147 HVGs identified using the variance decomposition-based method in at least one vascular region was performed based on NCBI PubMed citations listed under the entry for the *Mus musculus* reference sequence for each HVG in the NCBI Gene database (https://www.ncbi.nlm.nih.gov). The identified citations were manually examined for experimental evidence of direct regulation of SMC function, cell proliferation, cell migration, inflammation, stem cell properties and/or development of cardiovascular disease (Supplementary Data 4c).

**Co-expression analysis and combined gene expression scores.** To define co-expressed modules among HVGs detected in S+L+ cells, we used weighted correlation network analysis implemented in the R package WGCNA[60,61] v1.63. Specifically, we used blockwiseModules function with the following parameters: power = 3, TOMType = "unsigned", minModuleSize = 5, reassignThreshold = 0, mergeCutHeight = 0.25, with the soft-thresholding power chosen based on assessing the dependence between it, the scale-free network topology and mean connectivity based on plots produced by pickSoftThreshold. The identified modules, in particularly the core VSMC network (ME1) used for expression correlation analysis, were robust to parameter variation, such as using different soft-thresholding powers and signed instead of unsigned correlation networks. Network modules in Fig. 7d and Supplementary Fig. 7c, d were visualised in Cytoscape[62] v2.6.1, with edge thickness proportionate to WGCNA co-expression weight between respective gene pairs.

Expression of network modules and other gene sets presented in the paper was summarised by their first principal component defined across analysed cells. Summarised expression of ME1 is referred to as the cVSMC score. Since the signs of principal components are arbitrary, we additionally set the sign of summarised expression values such that they positively correlate with the mean expression values of the respective gene sets across cells.

**Detection of genes correlated with the cVSMC network.** To detect genes whose expression correlate (positively or negatively) with the cell's cVSMC network score (summarised expression of ME1, see above), we employed an approach taken for trajectory association analysis in the Bioconductor R package monocle[63], with some modifications. First, we estimated the mean-variance relationship based on all genes by fitting a robust local regression to the log-mean normalised counts versus log-squared coefficient of variation (log-SCV) using the loess function in R (family

= "symmetric"). The dependence of normalised expression counts for a given gene and cell on the cell's distance along the PC1 axis $d_t$ was then assessed by negative binomial (NB) regression implemented in the R package VGAM[64] v1.0-5, whereby the size parameter of the corresponding NB distribution was assumed to be the inverse fitted SCV obtained as described above. The significance of the regression fit was assessed by the likelihood-ratio test versus the intercept-only model and the resulting $p$-values were adjusted for multiple testing using the Benjamini-Hochberg procedure. Genes showing adjusted $p$-values below 0.05 were selected for further analysis.

**Analysis of the 10X Chromium single-cell RNA-seq data.** Raw sequencing reads were processed and aligned to the GRCm38 mouse genome through the 10X Genomics cellranger pipeline (v2.0 for VSMC-derived plaque cells and v2.1 for whole aorta cells and VSMC-lineage cells from healthy vessels). Quality control was performed based on total number of UMI counts per cell (atherosclerotic plaque dataset >5000 and <20,000, whole aorta/lineage-positive medial cells >1000 and <8000 UMI counts), genes detected per cell (atherosclerotic plaque dataset >1000 and <5000, whole aorta/lineage-positive medial cells >500 and <2500) and percentage of mitochondrial reads per cell (atherosclerotic plaque dataset <9%, whole aorta/lineage-positive medial cells <8%). Reads from cells passing the quality control criteria were normalised and scaled using the NormalizeData (scale.factor = 10,000) and ScaleData functions from the CRAN R package Seurat[65] v2.3.1. Principal component analysis was performed and the first five principal components used for clustering and t-distributed Stochastic Neighbour Embedding (t-SNE)[66] visualisation using the Seurat package. Differential gene expression between clusters was determined using the "tobit" method of the FindMarkers function implemented in Seurat ($p_{adj} < 0.05$). For identification of genes upregulated in individual VSMC clusters in the analysis of cells from the whole aorta, minimum log fold change of 0.25 and minimum percentage of expressing cells of 25% were required for identification as an upregulated gene.

**Gene ontology analysis.** Gene ontology (GO) analysis of genes differentially expressed between AA and DT cells was carried out using the online Functional Annotation Clustering tool in The Database for Annotation, Visualisation and Integrated Discovery (DAVID[67] v6.8) for all Biological Processes (BP) terms against all *Mus musculus* genes with medium classification stringency. Clusters were ranked by enrichment and fdr-adjusted $p$-values for the most significant term listed. GO enrichment analysis of genes positively and negatively correlated with cVSMC network expression was performed based on BP terms using clusterProfiler[68] package in R v3.8.1. KEGG pathways[40] in Supplementary Fig. 8 were visualised using the R package pathview[69] v1.20.

**Code availability.** R packages were downloaded from CRAN and Bioconductor. Custom code will be made available upon request.

## Data availability

Raw and processed sequencing data is available in the Gene Expression Omnibus (GEO) repository under accession number GSE117963. Previously published single-cell profiles of cultured VSMCs are available under accession number GSE79436.

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

## Acknowledgements

The authors would like to thank Paula Freire-Pritchett for help with data analysis, Simon Andrews, Steven Wingett, Felix Krueger at the Babraham Institute's Bioinformatics facility for help with data management and initial processing, Kristina Tabbada and Clare Murnane (Babraham Institute sequencing facility) for Illumina sequencing, Arthur Davis (Babraham Flow Facility), the Wellcome Trust-Medical Research Council, Institute of Metabolic Science, Metabolic Research Laboratories, Imaging core, Wellcome Trust Strategic Award [100574/Z/12/Z] for technical assistance and the Cambridge National Institute for Health Research Biomedical Research Centre Cell Phenotyping Hub for cell sorting, Alison Finigan (Cardiovascular Medicine Division, University of Cambridge) and Mark Lynch (Fluidigm) for technical assistance, Hashem Koohy for advice on random forest analysis and all members of the Spivakov and Jørgensen labs for helpful discussions. A.L.T., J.C. and J.L.H. are supported by British Heart Foundation (BHF) studentships (FS/15/62/32032, RE/13/6/30180, FS/15/38/31516) and L.D. and E.O. are supported by BBSRC DTP studentships. E.D. is supported by the ERC AdG 341096. H.F.J. and M.R.B. are supported by the BHF Centre of Regenerative Medicine (RM/13/3/30159), the BHF Cambridge Centre of Research Excellence (RE/13/6/30180) and a BHF Chair award (CH/20000003). M.S. is supported by core funding from the Medical Research Council of the UK.

## Author contributions

M.S. and H.F.J. designed the project; A.L.T., J.C., J.L.H. and H.F.J. performed the experiments; L.D., E.O., P.O., A.L.T., M.S. and H.F.J. analysed the data; M.S. and H.F.J. wrote the paper with contributions from all authors; M.R.B. and E.D. contributed reagents and analysis tools; M.S., H.F.J. and M.R.B. secured project funding; M.S. and H.F.J. supervised the work.

## Additional information

**Competing interests:** The authors declare no competing interests.

