## [Peer Review File · Nature Communications]

Reviewers' Comments:

Reviewer #1:

Remarks to the Author:

Vascular smooth muscle cell heterogeneity in healthy blood vessels revealed with single cell transcriptomics and lineage labelling

In their manuscript, Dobnikar et al seek to identify and describe the vSMC heterogeneity in healthy arteries. More specifically, they use single cell RNA-seq (scRNA-seq) combined with vSMC lineage tracing to describe the similarities and differences between cell subsets within healthy arteries. In this system, they were able to detect differences between vSMC from aortic arch that are of neural crest origin and the descending thoracic aorta that are derived from mesoderm. Of interest, they observed interesting region specific gene signatures that appeared to be a function of regional identity of the cells rather than the differential composition of cell populations between regions. Of even greater potential interest, authors detect a rare (<1%) population of SMC lineage+ (i.e. Myh11 expressing) cells that express the stem cell progenitor marker Sca1. They then perform a bioinformatics index sorting of these cells including three different populations: lineage tracing expressing Sca1 (S+L+), lineage tracing not expressing Sca1 (S-L+), and Sca1+ cells that did not express lineage tracing markers (S+L-). Using a diffusion component analysis, they showed that they could segregate contractile vSMCs (with or without Sca1 expression), from cells with a different phenotype. With further bioinformatics analysis, they suggest that there is a range of phenotypes between mature/contractile vSMCs to endothelial cell-like phenotypes in a healthy artery. Finally, although really emphasized, authors provide valuable data showing that gene signatures of cultured SMC are profoundly different from any of the single cell gene signatures of SMC in vivo. This is important since numerous studies in the field have placed far too much emphasis on results observed in cultured SMC without sufficient validation in vivo. Overall, although the paper is descriptive rather than mechanistic, results are of major value and potential impact for the vascular biology field. I have only several minor suggestions for revision.

Minor (?) Concerns:

1. The paper is very descriptive. This is expected from a heavy bioinformatics approach. However, it would be beneficial if the authors included a figure of a suggested model/mechanism to be tested in the future. There are several suggestions of what is happening in a healthy tissue in the discussion, but no proposed mechanism.
2. RNA-seq (bulk and single cell) alignment was performed using Tophat. Even though popular and still used, this software has entered low maintenance/support in 2016 and has been superseded by HISAT2 (<http://ccb.jhu.edu/software/hisat2/index.shtml>, from the same group). No big differences are expected, but it would be to the authors advantage to use a more updated and supported software (with more functionality as well).
3. Authors should tone down their speculative statements regarding how the observed heterogeneities between SMC in different regions influences disease susceptibility. While this is certainly possible, much more extensive mechanistic studies will be needed to show this is the case. Similarly, they should tone down their conclusions regarding the role of regional versus environmental effects on SMC gene signatures in that they have not directly isolated these as experimental variables and have only examined two out of what is believed to be eight distinct embryological sources of vascular SMC. I am not proposing that they do these experiments but rather they tone down their conclusions regarding these data throughout.
4. Authors have perhaps understated the significance of their single cell RNAseq comparisons of cultured SMC versus those identified in vivo. They conclude that data are consistent with cultured cells undergoing phenotypic changes but it is not clear they are mimicking a SMC state ever seen in vivo. Consistent with this idea, previous genome wide epigenetic profiling by Brad Bernstein and co-workers (Cell 2006) demonstrated that different cell types in vivo each have a distinct epigenomic signature based on principle component analyses. However, when placed in culture they acquired a common genome wide epigenomic signature irrespective of their origins. This suggests that cell culture had a dominant effect inducing a distinct signature that may not have an

in vivo counterpart. The limitation of these epigenomic studies is the lack of single cell resolution. However, it would seem the authors have the data to address this issue as a major conclusion of their study.

Reviewer #2:

Remarks to the Author:

In this study the authors studied the heterogeneity of healthy aortic media using the single cell transcriptomics and lineage mapping. One main finding is the presence of the Sca1-expressing cells that have both VSMC and EC phenotypes and gene signatures. The authors suggested that this intermediate cell population represents a functional VSMC heterogeneity and plasticity, which may be related to vascular disease. However, because the Sca1-expressing cells are rare in the aortic media, but are the major population within the adventitia, and perhaps in the endothelial layer too, more definitive experiments are needed to rule out the potential contamination and/or confirm the identify and biology significance of these Sca1-expressing cells.

Major concerns:

The authors carried out two sets of single cell transcriptomics analysis. The first used an unbiased approach for 143 cells isolated from the aortic media, and this identified one Sca1-expressing cell that expresses major VSMC markers. The second approach used cell sorting with Sca1 staining and VSMC lineage marker (Myh11-expressing and daughter cells) that separated the medial cells into 3 groups, Sca1-negative/Lineage+ (definitive VSMC), Sca1-expressing/Lineage+ (most definitive VSMC), and Sca1-expressing/Lineage- (most likely to be ECs, but expressing some level of VSMC genes). For the first approach, the sample size was too small to identify and characterize the rare Sca1-expressing cells. The second approach relied on the specificity of the Sca1 antibodies. Because the two limitations and the rareness of the Sca1-expressing cells in the media of the aorta, a scaled up scRNA-seq on thousands of cells need to identify and characterize the Sca1-expressing cells. This scaled up unbiased approach can simultaneously analyze the adventitial, medial and endothelial layers, by clustering them according to gene expression signatures, to provide the location information and cellular identify.

The authors did not provide any lineage development and functional evidence of the intermediate Sca1-expressing cells, although the authors labeled and compared recombination efficiency at the stages between one week and 16 weeks of post-induction of recombination. It would be an important to provide a comparison of single cell transcriptomics between the stages. The definitive answer requires, though, the direct lineage tracing of the Sca1+ cells in the aorta.

Specific comments:

Fig. S5a,b: 0.5-1% of media cells is Sca1+. ~80% of adventitial cells is Sca1+. What % of ECs is Sca1+?

Fig. 5d,e: Mislabeled in the legends (b or c is d or e). 47% of media cells are EYFP+ (labeling efficiency is ~50%), thus VSMC lineage; 7.4% of EYFP+ cells are Sca1-expressing cells, or ~4% of medial VSMC express Sca1, considering the labeling efficiency. On the other hand, the majority (>90%) of Sca1-expressing cells in the medial layer of aorta do not express the VSMC lineage marker EYFP. Again, after considering the 50% of labeling efficiency and assuming the equal distribution of Sca1-expressing cells in EYFP+ and EYFP- VSMCs, these was still a significant portion of (~45%) Sca1+ medial cells of unknown identifies or they are not VSMCs. What are these cells? ECs or ECs within the adventitia were accidentally included due to contamination during isolating the medial layer? It would be necessary to sort the Sca1-expressing cells using an EC surface maker.

Fig. 6. DC1 population consists of all mature VSMCs expressing the lineage maker EYFP, and 75% of Sca1-expressing and EYFP-expressing VSMCs. Sca1-expressing cells negative for EYFP express a low level of VSMC genes, but a high level of EC genes. Invasive circulating blood cells into the media? The EMT by the ECs? One experiment to look into the endothelial identity of the Sca1-expression cells is sorting medial vs whole aortic cells using antibodies for CD31 and Sca1. Again, one important question that was not addressed is the fate change of these intermediate cells, with VSMC and EC gene signatures. This is essential to show a rare cell population having a significant biological function.

Reviewer #3:

Remarks to the Author:

In this manuscript Dobnikar et al. exam the heterogeneity of vascular smooth muscle cell (VSMCs) across and within two vascular regions: the athero-prone aortic arch (AA) and the descending thoracic aorta (DT) in healthy mouse. Using a combination of bulk and single cell RNA-seq data analysis, they identify a higher expression of immune response, cell proliferation and migration genes but a lower expression of developmental regulators in AA compared with DT VSMCs, which might reflect their region-specific developmental origin. In terms of VSMS heterogeneity within vascular regions, the authors focus on Sca-1 positive VSMS-lineage cells, and show that the Sca-1 positive VSMS-lineage cells can be mapped along an axis that connects conventional VSMSs with endothelial-like cells. Based on this, the authors suggest a vascular plasticity between the smooth muscle and endothelial cell states.

This is in general an interesting story. Although the cell heterogeneity among VSMCs is known before, it hasn't been investigated at the single cell level. The authors also properly applied two mouse models to demonstrate the cell heterogeneity of Sca-1 positive cells.

My main concern however is that the analysis performed in this study are mostly based on biased approaches. For instance, the analysis of VSMC heterogeneity between AA and DT is based on a limited number of highly variable genes (HVGs), and the analysis of cell heterogeneity within vascular regions is focused on Sca-1 positive cells. Would be nice to see an unbiased cell type identification and cellular heterogeneity study both across and within vascular regions. To perform that analysis, the authors might consider to increase the cell number applied to the single cell RNA-seq.

Despite the new single cell RNA-seq methods applied, most of the conclusions in this paper are already known. To strengthen the significance and novelty of this work, the authors are highly suggested to perform further mechanistic studies to explore some of their key discoveries, such as the endothelial property of the Sca-1 positive VSMCs lineage cells. This can be easily done by an in vitro experimental verification using sorted cells.

We are pleased that the Reviewers found the work interesting and thank them for their comprehensive and balanced comments. In response to their feedback, we have increased the number of cells analysed, including unbiased analyses of thousands of cells, added experimental evidence for the functional relevance of the identified Sca1+ population and significantly revised the manuscript.

The revised manuscript presents a comprehensive analysis of VSMC heterogeneity in healthy vessels, including evidence for a rare population of Sca1+ VSMC-lineage cells with a likely role in VSMC response. In addition, our study provides a vast resource of single-cell RNA-seq profiles of cells from healthy vessels and atherosclerotic plaques, which in our view will be of high interest to the research community.

We believe that the study has been significantly improved by these changes, which are summarised below and detailed in the point-by-point response to individual Reviewers (in blue).

- We now complement the focused analyses of specific VSMC subpopulations in healthy vessels with lower-coverage profiles of thousands of unselected cells from the whole aorta, as well as VSMC-lineage cells. Here, we present these datasets in the context of further validating the HVGs detected with higher-coverage data from smaller cell numbers, and, in particular, to confirm the Sca1+ medial subpopulation. However, we release these data in full and believe that it will provide a valuable resource for other analyses by the vascular research and single-cell genomics communities.
- We further validate the Sca1+ medial subpopulation in healthy cells using Sca1-GFP transgenic animals, circumventing the need for Sca1 antibody.
- We have increased the number of cells analysed by Smart-seq2 and retrospectively implemented a more stringent gating strategy to select VSMC-lineage positive cells. The revised manuscript presents the analysis of 92 index-sorted Sca1+ VSMC-lineage (S+L+) cells.
- Based on the results of the revised analyses, we propose that Sca1 upregulation reflects an activated, plastic cell state that is likely relevant for VSMC phenotypic switching in disease. We substantiate this hypothesis, now outlined in Figure 10, with functional evidence. Specifically, we demonstrate that Sca1 is upregulated in VSMCs *in vitro*, as well as during *in vivo* injury-induced VSMC phenotypic switching (new Figure 8). We further show that VSMC-lineage Sca1+ cells represent a significant subpopulation of cells in atherosclerotic plaques isolated from ApoE^{-/-} animals on a cholesterol-rich diet. We also show that Sca1-positive VSMC-derived plaque cells express the transcriptional signatures identified in their counterparts in healthy vessels (new Figure 9). These analyses provide evidence for a functional role of S+L+ cells in VSMC phenotypic modulation and open up the possibility that the frequency of S+L+ cells in healthy vessels and/or their position on the axis defined by core

VSMC genes may reflect the 'priming' of the vasculature for inflammatory response.

- We accept Reviewer 2's criticism that the identity of Sca1-positive, VSMC lineage-negative (S+L-) cells is unclear. We therefore no longer rely on these cells for identifying the transcriptional signatures of Sca1+ medial cells, and only use S+L+ cells in this analysis. For the same reason, we have redefined the axis along which we profile the heterogeneity of these cells, such that it is now based entirely and explicitly on the markers of conventional VSMCs. The increased stringency and power of the analysis has allowed a robust identification of the transcriptional signatures associated with VSMC phenotypic response in S+L+ cells, leading to further insights regarding their function (new Figure 7).
- We take on board the Reviewers' concern that the endothelial plasticity of VSMC-lineage cells requires further functional validation. However, we found that the very small numbers of S+L+ cells and their significant heterogeneity make experiments required to validate this phenomenon technically challenging. We have therefore toned-down claims to this extent in the manuscript.

Reviewers' comments:

Reviewer #1 - expert in SMCs (Remarks to the Author):

Vascular smooth muscle cell heterogeneity in healthy blood vessels revealed with single cell transcriptomics and lineage labelling

In their manuscript, Dobnikar et al seek to identify and describe the vSMC heterogeneity in healthy arteries. More specifically, they use single cell RNA-seq (scRNA-seq) combined with vSMC lineage tracing to describe the similarities and differences between cell subsets within healthy arteries. In this system, they were able to detect differences between vSMC from aortic arch that are of neural crest origin and the descending thoracic aorta that are derived from mesoderm. Of interest, they observed interesting region specific gene signatures that appeared to be a function of regional identity of the cells rather than the differential composition of cell populations between regions. Of even greater potential interest, authors detect a rare (<1%) population of SMC lineage+ (i.e. Myh11 expressing) cells that express the stem cell progenitor marker Sca1. They then perform a bioinformatics index sorting of these cells including three different populations: lineage tracing expressing Sca1 (S+L+), lineage tracing not expressing Sca1 (S-L+), and Sca1+ cells that did not express lineage tracing markers (S+L-). Using a diffusion component analysis, they showed that they could segregate contractile vSMCs (with or without Sca1 expression), from cells with a different phenotype. With further bioinformatics analysis, they suggest that there is a range of phenotypes between mature/contractile vSMCs to endothelial cell-like phenotypes in a healthy artery. Finally, although really emphasized, authors provide valuable data showing that gene signatures of cultured SMC are profoundly different from any of the single cell

gene signatures of SMC *in vivo*. This is important since numerous studies in the field have placed far too much emphasis on results observed in cultured SMC without sufficient validation *in vivo*. Overall, although the paper is descriptive rather than mechanistic, results are of major value and potential impact for the vascular biology field. I have only several minor suggestions for revision.

We thank the Reviewer for the positive evaluation of our manuscript and provide responses to their specific concerns below.

Minor (?) Concerns:

1. The paper is very descriptive. This is expected from a heavy bioinformatics approach. However, it would be beneficial if the authors included a figure of a suggested model/mechanism to be tested in the future. There are several suggestions of what is happening in a healthy tissue in the discussion, but no proposed mechanism.

We appreciate the Reviewer's understanding that the approach taken in our paper - the in-depth characterisation of VSMC populations, primarily in healthy tissue - inevitably leads to findings that may be considered descriptive. The revised version now includes new analyses demonstrating that subsets of VSMCs activate Sca1 upon vascular injury and in atherosclerotic plaques, which add functional insights.

As suggested by the Reviewer, we include a figure (Figure 10) presenting a model for the role of Sca1+ VSMC-lineage cells in phenotypic switching. This model is directly testable and will need to be validated experimentally in the future, pending the development of specific lineage-tracing tools required for this analysis.

2. RNA-seq (bulk and single cell) alignment was performed using Tophat. Even though popular and still used, this software has entered low maintenance/support in 2016 and has been superseded by HISAT2 (<http://ccb.jhu.edu/software/hisat2/index.shtml>, from the same group). No big differences are expected, but it would be to the authors advantage to use a more updated and supported software (with more functionality as well).

To assess whether the choice of the algorithm affects our results, we compared HISAT2 with Tophat alignment on a subset of single-cell profiles generated using Fluidigm C1 (85 cells) and Smart-seq2 platforms (81 cells). Correlation plots for mean expression levels per gene (Reviewer Figure 1) and expression levels of selected genes per cell (Reviewer Figure 2) for data aligned with Tophat and HISAT2 are shown below and suggest that, for our data, HISAT2 and Tophat yield generally consistent results. Importantly, several key results in our study were verified using the newly added high-throughput 10X Chromium single-cell data, which were processed using another aligner (STAR) as part of the 10X data processing pipeline, providing further confidence that the conclusions are independent of the choice of aligner. We therefore do not feel that swapping the aligner and performing a full reanalysis of a dataset of this size and complexity would be needed.

Reviewer Figure 1: Comparison of Tophat and Hisat2 for mean gene expression levels. Correlation plots showing mean expression levels in Fluidigm C1 (top panels) and Smart-seq2 (lower panels) experiments calculated after alignment using Hisat2 (plotted along the X axis) versus Tophat (Y axis). Each dot shows the mean expression of one gene across cells (85 cells for Fluidigm C1 and 81 cells for Smart-seq2 experiments). Panels at the right show magnified views of lowly expressed genes (mean expression 0-5000).

Reviewer Figure 2: Comparison of Hisat2 and Tophat-aligned gene level data. Correlation plots showing expression levels for indicated genes in Fluidigm C1 (top panels) and Smart-seq2 (lower panels) experiments calculated after alignment using Hisat2 (X axis) versus Tophat (Y axis). Dots show the read count for the indicated gene in each single cell (85 cells for Fluidigm C1 and 81 cells for Smart-seq2 experiments).

3. Authors should tone down their speculative statements regarding how the observed heterogeneities between SMC in different regions influences disease susceptibility. While this is certainly possible, much more extensive mechanistic studies will be needed to show this is the case. Similarly, they should tone down their conclusions regarding the role of regional versus environmental effects on SMC gene signatures in that they have not directly isolated these as experimental variables and have only examined two out of what is believed to be eight distinct embryological sources of vascular SMC. I am not proposing that they do these

experiments but rather they tone down their conclusions regarding these data throughout.

We agree that a formal validation of the role of the identified SMC heterogeneity and embryonic origin on disease susceptibility is still required and goes beyond the scope of this study. We have therefore toned-down the discussion of these points in the text [page 5 (end of 2nd paragraph deleted), page 5 last paragraph and page 12/13].

4. Authors have perhaps understated the significance of their single cell RNAseq comparisons of cultured SMC versus those identified *in vivo*. They conclude that data are consistent with cultured cells undergoing phenotypic changes but it is not clear they are mimicking a SMC state ever seen *in vivo*. Consistent with this idea, previous genome wide epigenetic profiling by Brad Bernstein and co-workers (Cell 2006) demonstrated that different cell types *in vivo* each have a distinct epigenomic signature based on principle component analyses. However, when placed in culture they acquired a common genome wide epigenomic signature irrespective of their origins. This suggests that cell culture had a dominant effect inducing a distinct signature that may not have an *in vivo* counterpart. The limitation of these epigenomic studies is the lack of single cell resolution. However, it would seem the authors have the data to address this issue as a major conclusion of their study.

We agree with the Reviewer that cultured cells provide a limited model of phenotypic VSMC modulation observed *in vivo* and have indicated this in the revised manuscript (page 3, 2nd paragraph). We used data from cultured cells as an illustration of the pronounced differences between the expression profiles of *in vitro* cultured and *ex vivo* derived VSMCs, particularly with respect to contractile VSMC genes. We now also use cultured cells to show that VSMCs upregulate Sca1 *in vitro* (page 10, middle and Figure 8a,b), as a stepping stone to the *in vivo* analyses presented later in the same section.

To what extent the transcriptional profile of cultured VSMCs reflects the “synthetic” versus “cell culture” signature is indeed an interesting question, which we explored using work from the Bernstein lab, as suggested (we believe the relevant reference is Zhu et al, Cell 2013). Unfortunately, this study has highlighted culture-associated signatures of chromatin repression (and in particular, H3K9 trimethylation), which cannot be directly compared with the transcriptional profiles we have generated. Indeed, there was very little overlap between the genes that we find to be differentially expressed in cultured versus *ex vivo* cells and those found to be differentially repressed by H3K9me3 in the Bernstein lab study. This was true both at the gene level (Reviewer Figure 3) and at the level of differentially enriched GO-terms identified in PAGODA analysis (data not shown).

Reviewer Figure 3: Comparison of culture-induced genes with culture-associated epigenetic regulation.

Top panels show Venn diagrams representing the overlap between the genes located within H3K9me3 domains associated with cultured cells (Zhu et al, 2013) and those we have identified (using scde) as upregulated in either the cultured (left) or ex vivo (right) single cells. Lower panels show the same overlap separately for each cluster of genes associated with H3K9me3 domains in cultured cells defined in Zhu et al, 2013, Figure 6B.

Reviewer #2 - expert in lineage tracing (Remarks to the Author):

In this study the authors studied the heterogeneity of healthy aortic media using the single cell transcriptomics and lineage mapping. One main finding is the presence of the Sca1-expressing cells that have both VSMC and EC phenotypes and gene signatures. The authors suggested that this intermediate cell population represents a functional VSMC heterogeneity and plasticity, which may be related to vascular disease. However, because the Sca1-expressing cells are rare in the aortic media, but are the major population within the adventitia, and perhaps in the endothelial layer too, more definitive experiments are needed to rule out the potential contamination and/or confirm the identify and biology significance of these Sca1-expressing cells.

Cell identity is indeed an important point that requires careful validation. In addition to using genetic lineage labelling, we have now performed a number of additional analyses to confirm that the identified VSMC-lineage Sca1-expressing cells are distinct from Sca1-expressing cells residing in the adjacent cell layers, as detailed in the responses to the Reviewer's specific concerns below.

To stay on the safe side, we also retrospectively tightened the FACS gate settings for the Confetti label used for lineage labelling, excluding an equal number of S-L+ and S+L+ cells (17 and 20, respectively). We also performed additional experiments using the stringent gating strategy in order to increase the number of cells analysed in the revised manuscript (in total of 92 S+L+ and 36 control S-L+ cells).

To demonstrate the functional significance of VSMC-lineage Sca1+ cells, we now provide evidence from both *in vitro* and *in vivo* experiments in healthy tissue and disease models:

- 1) We show evidence that VSMCs upregulate Sca1 in culture and *in vivo*, both in healthy tissue and upon vascular injury (Figure 8).
- 2) We used 10X Chromium technology to profile VSMC-lineage cells sorted from the atherosclerotic plaques of ApoE^{-/-} mice fed a cholesterol-rich diet (Figure 9a, ~3300 cells). This analysis revealed a sizeable cluster of Sca1-expressing VSMC-derived cells in the plaque (Figure 9b). We further show that Sca1-positive plaque cells are enriched for the "VSMC Response Signature" (Figure 9d, cVSMCneg genes) we identified in S+L+ cells from healthy vessels, which is expressed at increased levels in cells where the cVSMC Signature (cVSMCpos genes) is reduced.

Based on the functional analyses and the characteristic expression signatures of S+L+ cells, we now propose a model for the functional role of Sca1-positive VSMCs in phenotypic response (Figure 10). This model opens the possibility that this subpopulation of non-conventional VSMC found in healthy tissue could be a therapeutic target and that the frequency of these cells within the vessel wall could reveal disease susceptibility.

Major concerns:

The authors carried out two sets of single cell transcriptomics analysis. The first used an unbiased approach for 143 cells isolated from the aortic media, and this identified one Sca1-expressing cell that expresses major VSMC markers. The second approach used cell sorting with Sca1 staining and VSMC lineage marker (Myh11-expressing and daughter cells) that separated the medial cells into 3 groups, Sca1-negative/Lineage+ (definitive VSMC), Sca1-expressing/Lineage+ (most definitive VSMC), and Sca1-expressing/Lineage- (most likely to be ECs, but expressing some level of VSMC genes). For the first approach, the sample size was too small to identify and characterize the rare Sca1-expressing cells. The second approach relied on the specificity of the Sca1 antibodies. Because the two limitations and the rareness of the Sca1-expressing cells in the media of the aorta, a scaled up scRNA-seq on thousands of cells need to identify and characterize the Sca1-expressing cells. This scaled up unbiased approach can simultaneously analyze the adventitial, medial and endothelial layers, by clustering them according to gene expression signatures, to provide the location information and cellular identify.

We have taken a number of steps to address these concerns:

- 1) We followed the Reviewer's suggestion and analysed unselected cells from whole aorta preparations (10X Chromium scRNA-seq analysis of ~2800 cells), which revealed a rare population of medial *Ly6a/Sca1*⁺ cells co-clustering with VSMCs rather than other Sca1-positive endothelial and adventitial cells (Figure 5a). We also detected *Ly6a/Sca1* transcripts in a 10X Chromium analysis of ~2800 sorted VSMC-lineage positive cells (Figure S6).

- 2) We find that both S+L+ and S-L+ cells express significant levels of genes showing selective expression in VSMCs in the whole aorta 10X Chromium dataset (Reviewer Figure 4 below). In contrast, genes that are selectively enriched in the endothelial and adventitial populations in this dataset (including the classic endothelial and adventitial markers Chd5, Vwf and Lum) are only sporadically expressed in S+L+ cells.
- 3) We also took advantage of transgenic Sca1-GFP mice to confirm that Sca1-positive cells in the medial layer (detected without relying on the specificity of Sca1 antibodies) express the VSMC marker α SMA, which is not expressed by adventitial and endothelial cells (Figure 5b,c).
- 4) For VSMC lineage-negative Sca1-expressing cells (S+L-), it is more difficult to rule out potential contamination from the neighbouring tissues, since (1) no lineage labelling is used, and (2) the expression profiles of these cells with respect to EC and adventitial markers do not preclude this possibility for at least some S+L- cells (Reviewer Figure 4). We have therefore refocused the analysis of the Sca1+ population to S+L+ cells only. We also redefine the axis, along which we arrange the cells, in a way that exclusively relies on the expression of a network of contractile VSMC genes. The gene signature identified in the new analysis are also expressed in plaque cells, providing functional relevance of the cells. Several endothelial genes were also detected in this analysis, consistent with our findings in the original submission. The endothelial plasticity of VSMCs therefore remains a possibility, but we agree that it needs to be further validated using alternative approaches and functional analyses. We have therefore toned-down this point in the revised version of the manuscript.

Reviewer Figure 4: S+L+ cells express VSMC markers.

Heatmap presenting the expression levels of the genes that showed selective expression in VSMC (yellow side bar), adventitial (green) or endothelial (red) cell clusters in the 10X Chromium analysis of the whole aorta (Manuscript Figure 5d). Population-selective marker genes are the top 20 genes showing differential expression between the clusters (tobit test adjusted p -value < 0.05) and expressed in >25% of the cells within the enriched cluster, ranked by log-fold change. VSMC markers are uniformly expressed in S-L+ (horizontal magenta bar) and S+L+ (yellow) cells, and both cell

populations show only sporadic expression of individual Adventitial and EC markers. In contrast, individual S+L- cells (turquoise horizontal bar) showed a uniform expression of either VSMC, EC or adventitial markers.

The authors did not provide any lineage development and functional evidence of the intermediate Sca1-expressing cells, although the authors labeled and compared recombination efficiency at the stages between one week and 16 weeks of post-induction of recombination. It would be an important to provide a comparison of single cell transcriptomics between the stages. The definitive answer requires, though, the direct lineage tracing of the Sca1+ cells in the aorta.

To assess lineage development, we have analysed the frequency of VSMC-lineage cells that express Sca1 at different times after tamoxifen-mediated recombination (Figure 8e). This analysis demonstrated a significant increase in the proportion of VSMC-lineage cells that are Sca1-positive over time, and that could not be explained away by age. This suggests that L+ cells upregulate Sca1 in healthy vessels, albeit at a low rate. Differential expression analysis of S+L+ cells isolated from animals sacrificed 2 or 13.5 weeks post tamoxifen treatment revealed little evidence of lineage development of these cells. While we did detect differential expression of a small number of genes (6 upregulated at 2 weeks and 77 genes upregulated at 13.5 weeks, Reviewer Figure 5), they did not show specific enrichment for any GO-terms or pathways.

Reviewer Figure 5: Labelling time-dependent differential expression in S+L+ cells.

Box plots showing expression levels (log-normalised counts) of genes detected as differentially expressed in S+L+ cells isolated from animals sacrificed two (orange bars) or 13.5 weeks (turquoise bars) after tamoxifen-mediated VSMC lineage labelling of Myh11-CreERT2/Confetti+ animals. The results are based on analysis of 15 (2 weeks) and 35 cells (13.5 weeks) from two separate samples per time point (3-4 animals per sample). Left panel shows genes upregulated in the animals analysed after 2 weeks and right panel shows top 20/77 genes upregulated in animals analysed after 13.5 weeks (based on scde, corrected Z-score > 1.96).

We now provide *in vivo* evidence from cardiovascular disease models [carotid injury (Figure 8f-h) and atherosclerotic plaques (Figure 9)] for the functional relevance of S+L+ cells in VSMC activation during phenotypic switching. We agree that selective lineage tracing of S+L+ cells is needed to provide definitive evidence to this extent, but this experiment would require the development of a dual recombinase/reporter system which is beyond the scope feasible for revisions of this study.

Specific comments:

Fig. S5a,b: 0.5-1% of media cells is Sca1+. ~80% of adventitial cells is Sca1+. What % of ECs is Sca1+?

In the 10X Chromium analysis of unselected cells, 63% of cells in the EC population are Sca1+, which is most likely an underestimate due to the low coverage of the 10X method. We therefore compared this result to flow cytometry analysis of the cell preparation of aortas, where the adventitial layer, but not ECs, was removed. We considered only VSMC lineage-negative cells and found that 4-8% of them expressed CD31 (not shown). Almost all (90-97%) ECs (CD31+ VSMC lineage-negative) expressed Sca1 (Reviewer Figure 6 below). For comparison, 44% of cells in the adventitial cell population identified in the 10X analysis had detectable levels of *Ly6a/Sca1*, versus 80% detected by antibody staining and flow cytometry.

Reviewer Figure 6: Cd31-positive cells express Sca1.

Aortas from tamoxifen-injected Myh11-CreERT2/EYFP animals (VSMC-specific expression of YFP) were dissected free of fat and connective tissue, pre-digested to remove the adventitia, but without manual removal of the intima, and digested to a single-cell suspension. Cells were stained for CD31 (BUV395) and Sca1 (APC), or isotype controls, and analysed by flow cytometry. Graphs show the percentage of YFP-negative, CD31+ cells that express Sca1. N = 3 animals.

Fig. 5d,e: Mislabeled in the legends (b or c is d or e).

Figure 5 has been changed, but labelling of figure panels has been corrected (now part of Figure 6).

47% of media cells are EYFP+ (labeling efficiency is ~50%), thus VSMC lineage; 7.4% of EYFP+ cells are Sca1-expressing cells, or ~4% of medial VSMC express Sca1, considering the labeling efficiency. On the other hand, the majority (>90%) of Sca1-expressing cells in the medial layer of aorta do not express the VSMC lineage marker EYFP. Again, after considering the 50% of labeling efficiency and assuming the equal distribution of Sca1-expressing cells in EYFP+ and EYFP- VSMCs, there was still a significant portion of (~45%) Sca1+ medial cells of unknown identities or they are not VSMCs. What are these cells? ECs or ECs within the adventitia were accidentally included due to contamination during isolating the medial layer? It would be necessary to sort the Sca1-expressing cells using an EC surface marker.

The identity of the S+L- cells is indeed unclear. We expect that this cell population contains a mix of medial non-VSMC-lineage cells that express Sca1, a minority of Sca1+ VSMCs that did not recombine the lineage label, and possibly contaminating

cells from adjacent cell layers (see Reviewer Figure 4). Owing to the heterogeneity of this population and the focus of our study of VSMC-lineage cells, we have decided to remove S+L- cells from the analyses in Figure 7 and beyond, as detailed above.

Fig. 6. DC1 population consists of all mature VSMCs expressing the lineage maker EYFP, and 75% of Sca1-expressing and EYFP-expressing VSMCs. Sca1-expressing cells negative for EYFP express a low level of VSMC genes, but a high level of EC genes. Invasive circulating blood cells into the media? The EMT by the ECs? One experiment to look into the endothelial identity of the Sca1-expression cells is sorting medial vs whole aortic cells using antibodies for CD31 and Sca1. Again, one important question that was not addressed is the fate change of these intermediate cells, with VSMC and EC gene signatures. This is essential to show a rare cell population having a significant biological function.

We agree that the origin and properties of medial Sca1-positive cells not expressing the lineage label is an exciting question. However, in view of the considerations presented above, we have decided to leave these cells out of detailed analysis in the present study. Their future investigation will require other lineage labelling models and functional experiments that are more appropriate for these cells.

Reviewer #3 - expert in single-cell RNA sequencing (Remarks to the Author):

In this manuscript Dobnikar et al. exam the heterogeneity of vascular smooth muscle cell (VSMCs) across and within two vascular regions: the athero-prone aortic arch (AA) and the descending thoracic aorta (DT) in healthy mouse. Using a combination of bulk and single cell RNA-seq data analysis, they identify a higher expression of immune response, cell proliferation and migration genes but a lower expression of developmental regulators in AA compared with DT VSMCs, which might reflect their region-specific developmental origin. In terms of VSMS heterogeneity within vascular regions, the authors focus on Sca-1 positive VSMS-lineage cells, and show that the Sca-1 positive VSMS-lineage cells can be mapped alone an axis that connects conventional VSMSs with endothelial-like cells. Based on this, the authors suggest a vascular plasticity between the smooth muscle and endothelial cell states.

This is in general an interesting story. Although the cell heterogeneity among VSMCs is known before, it hasn't been investigated at the single cell level. The authors also properly applied two mouse models to demonstrate the cell heterogeneity of Sca-1 positive cells.

We are happy that the Reviewer appreciates the interest of our study and relevance of the model systems we used.

My main concern however is that the analysis performed in this study are mostly based on biased approaches. For instance, the analysis of VSMC heterogeneity between AA and DT is based on a limited number of highly variable genes (HVGs), and the analysis of cell heterogeneity within vascular regions is focused on Sca-1

positive cells. Would be nice to see an unbiased cell type identification and cellular heterogeneity study both across and within vascular regions. To perform that analysis, the authors might consider to increase the cell number applied to the single cell RNA-seq.

As requested by the Reviewer, we have added unbiased analyses to complement and test our conclusions. In particular, we now take advantage of 10X Chromium technology to additionally profile thousands of cells, albeit at a lower coverage than in Fluidigm C1 and Smart-seq2 experiments. We profile ~2800 unselected aortal cells to provide an unbiased cell type identification (Figure 4d). Remarkably, we show that 13% of the HVGs identified in the Fluidigm-based analysis of AA and DT cells are differentially expressed between distinct subpopulations of VSMCs detected in the 10X Chromium experiment (see p. 6, 3rd paragraph and Figure 4e), underscoring the robustness of our findings regarding VSMC heterogeneity across different platforms, numbers of profiled cells and sequencing coverage. In addition, we use 10X Chromium technology to specifically characterise the heterogeneity of VSMC-lineage cells across thousands of cells in the healthy tissue and the plaque (Figure S6a and Figure 9).

We believe that there was a slight misunderstanding regarding our analysis methodology, which we hope is clarified in the revised manuscript. In fact, the analysis of heterogeneity between vascular regions is based on genes showing regional expression in the bulk RNA-seq analysis rather than HVGs. Given that bulk RNA-seq generally has a higher sensitivity, we favour this approach to reduce the chance of including genes with spurious differences in detection rates between the two populations in the analysis. However, we repeated the random forest analysis without restricting the gene list during model training, as requested by the Reviewer. This results in an area under the curve of 89% (Reviewer Figure 7), which is only slightly lower than that in the model based on only the differentially expressed genes detected in the bulk RNA-seq dataset.

Reviewer Figure 7: ROC plot of Random Forest analysis based on all detected protein coding genes.

ROC curve showing the accuracy of a random forest classifier that was trained based on the expression of all genes detected within 75% of cells (the same training set as used in the manuscript), when applied to the 25% of cells that were not used for training.

Despite the new single cell RNA-seq methods applied, most of the conclusions in this paper are already known. To strengthen the significance and novelty of this work, the authors are highly suggested to perform further mechanistic studies to explore some of their key discoveries, such as the endothelial property of the Sca-1 positive VSMCs lineage cells. This can be easily done by an *in vitro* experimental verification using sorted cells.

As mentioned above, we have refocused the second part of the paper based on our finding of a "VSMC Response Signature" that is progressively upregulated in S+L+ cells from healthy vessels (Figure 7), which we believe is a novel and exciting finding. As suggested, we follow this up with a number of functional analyses supporting the role of S+L+ cells in phenotypic switching. In particular, we provide evidence that contractile VSMCs activate Sca1 expression in culture (Figure 8a-d) and *in vivo* in both healthy tissue (Figure 8e) and upon carotid ligation (Figure 8f-h). We also use 10X Chromium technology to profile VSMC-lineage cells from atherosclerotic plaques isolated from ApoE^{-/-} mice fed a cholesterol-rich diet, revealing a significant subpopulation of S+L+ cells that express the transcriptional signatures of their counterparts from healthy vessels (Figure 9).

We also confirm increased expression of several EC-associated genes (e.g., *Vcam1* and *Ft1*) in S+L+ cells, which may be indicative of plasticity towards the EC lineage. The ability of Sca1+ medial cells to respond to high doses of signals that induce both endothelial and VSMC markers has been reported previously (Sainz et al., *Arterioscler. Thromb. Vasc. Biol.* 2006, 26, 281–286). However, the specificity of this effect and its reproducibility in the biological doses of signals remains to be elucidated. Unfortunately, functional tests of the endothelial plasticity of S+L+ cells (by assays such as survival in EC culture conditions and acetylated LDL uptake) proved highly challenging due to the very small numbers of available S+L+ cells (100-150 per animal) and their considerable heterogeneity (as evident from Figure 7). We therefore accept that conclusive evidence of these cells' endothelial plasticity is still lacking and have toned-down this aspect of our findings in the revised version of our study.

POINT-BY-POINT RESPONSE: (response in green font)

REVIEWERS' COMMENTS:

Reviewer #1 (Remarks to the Author):

The authors have fully addressed my previous concerns and this revised manuscript is much improved. I also like the substantial revisions and additional studies they have included in their response to the other reviewers.

Reviewer #2 (Remarks to the Author):

The authors have done an excellent job in revising the manuscripts. New data from additional scaled up single cell RNA-seq and functional experiments support the original, major conclusion. I appreciate the authors' effort that has addressed my previous questions. In addition, I agree with the authors that the origin(s) and development of the Sca1+ VSMCs are topics of future studies.

Reviewer #3 (Remarks to the Author):

In the revision manuscript, Dobnikar et al. have performed several experiments well addressed my concerns to the last version of manuscript. Also, the paper has restructured and reworded. Therefore, we believe this manuscript could now be considered for publication in Nature Communications.

We are pleased the reviewers agree that the revised version of our manuscript has addressed all concerns raised.